# SDAE: Semantic-Diversity-Aware Exploration for Efficient Reinforcement Learning in Large Language Models

## Abstract

Reinforcement learning with verifiable rewards (RLVR) significantly enhances the reasoning capabilities of large language models. However, as training progresses, models often converge to narrow solution paths, leading to diversity collapse. Existing methods alleviate this phenomenon through token-level entropy regularization or negative sample reinforcement, yet these works have treated responses as atomic units distinguished only by binary correctness labels, overlooking substantial semantic heterogeneity among responses under the same label—among incorrect responses, the degree and type vary significantly; among correct responses, both conventional and novel strategies exist. Uniformly reinforcing all correct responses leads to policy convergence, while uniformly penalizing all incorrect ones may suppress promising directions. Building on this insight, we propose **S**emantic **D**iversity-**A**ware **E**xploration (**SDAE**), which leverages geometric structures in semantic space to guide exploration. SDAE computes the semantic centroid of positive samples and modulates advantage functions based on each response's distance to this centroid, encouraging novel correctness, penalizing divergent errors, while preserving improvement potential in locally flawed responses. Experiments demonstrate that SDAE consistently outperforms strong baselines across competition-level reasoning benchmarks, achieving a 13.3% improvement in Pass@256 on AIME25. Our code is available at https://anonymous.4open.scienc e/r/SDAEcode-3553.

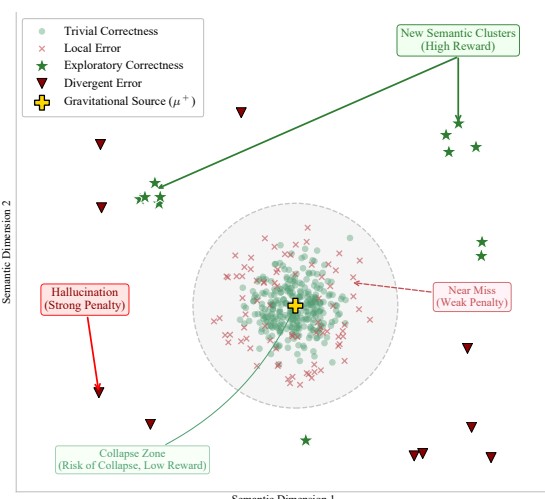

*Figure 1.* The Semantic Geometry of Exploration. This figure illustrates how SDAE categorizes and modulates responses within the latent manifold of the reasoning space.

## 1. Introduction

Reinforcement learning with verifiable rewards (RLVR) has emerged as the dominant paradigm for enhancing reasoning capabilities in large language models (LLMs) (Guo et al., 2025; Shao et al., 2024; Wang et al., 2025b), with its core mechanism reinforcing correct responses and penalizing incorrect ones through policy gradient algorithms in a simple yet effective manner (Schulman et al., 2017; Shao et al., 2024). However, this purely accuracy-driven optimization objective harbors a fundamental tension: as training progresses, models tend to converge toward narrow solution paths, gradually losing the impetus to explore alternative reasoning strategies. This phenomenon, commonly referred to as diversity collapse, manifests as a sustained decline in the entropy of output distributions and stagnation or even degradation in performance on complex reasoning tasks (Dang et al., 2025; Yue et al., 2025; Cui et al., 2025b).

In traditional reinforcement learning, entropy serves as a fundamental metric for exploration, quantifying the uncertainty in a model's action distribution. To address diversity collapse, existing research has primarily proceeded along two lines. The first line follows maximum entropy reinforce-

---

[1]Anonymous Institution, Anonymous City, Anonymous Region, Anonymous Country. Correspondence to: Anonymous Author <anon.email@domain.com>.

Preliminary work. Under review by the International Conference on Machine Learning (ICML). Do not distribute.

ment learning principles, maintaining exploration capability by encouraging policy entropy (Haarnoja et al., 2018; Ziebart et al., 2008). Recent work has demonstrated positive correlations between high-entropy regions and exploratory reasoning actions (Cheng et al., 2025; Wang et al., 2025a). The fundamental limitation of such methods lies in their operation at the token level, measuring uncertainty in lexical choices rather than semantic content—models may exhibit high entropy through surface phrasing variations while repeatedly exploring the same underlying strategies. The second line employs negative sample reinforcement (NSR), penalizing only incorrect responses to maintain generation diversity (Zhu et al., 2025; Tang et al., 2025). NSR effectively prevents model overconfidence on correct samples by suppressing erroneous generations and redistributing probability mass to reasonable alternatives. However, NSR treats all incorrect responses uniformly, regardless of whether they represent promising near-miss directions or fundamental errors, while also failing to actively guide models toward exploring semantically diverse correct solutions.

Through in-depth analysis of the training process, we observe a critical granularity deficiency in how existing methods handle response "quality." Taking mathematical reasoning as an example: among incorrect responses, there exist "near-miss" local failures and "completely off-track" fundamental errors; among correct responses, there exist "clichéd" conventional paths and "innovative" novel strategies. The binary labels universally adopted by existing methods obliterate these crucial nuances, failing to identify "near-miss" samples with improvement potential or to provide additional incentives for genuinely exploration-worthy novel solutions. This signal deficiency is precisely what causes models to fall into diversity collapse during late training stages, struggling to break through reasoning bottlenecks.

We discover that this hidden quality differential exhibits remarkably clear geometric manifestation in semantic embedding spaces. As illustrated in Figure 1, when mapping model responses to latent space manifolds, responses of varying qualities exhibit distinctive topological signatures on the latent manifold: semantically similar conventional solutions spontaneously cluster around the centroid of correct solutions, while novel solutions and divergent errors distribute in peripheral regions far from the centroid. This discovery reveals a key insight: **the geometric distance from a response to the semantic centroid essentially encodes its marginal value for model exploration.** Based on this, we construct a four-category semantic geometry framework: Trivial Correctness (conventional solutions), Exploratory Correctness (novel paths), Local Error (near-miss), and Divergent Error (fundamental failures). This perspective exposes the common blind spot of existing methods: neither token-level entropy nor binary labels exploit the geometric information embedded in semantic space.

Building on these insights, we propose Semantic-Diversity-Aware Exploration (SDAE), a framework that guides exploration through semantic space geometry rather than token-level entropy. Its core logic lies in using the semantic centroid of correct responses as a "gravitational source," measuring each response's distance to this centroid (termed Semantic Gravity), and differentially modulating advantage functions accordingly. Specifically, SDAE comprises three steps: first, leveraging the policy model's own hidden states to perform semantic aggregation on sampled responses, with the cluster center of correct responses serving as the gravitational source; second, computing each response's semantic gravity to the centroid and combining it with correctness labels to classify samples into four categories; finally, establishing tiered advantage modulation—amplifying rewards for novel correct paths, strengthening penalties for divergent errors, and attenuating penalties for local errors to preserve improvement potential. Experiments demonstrate that SDAE consistently outperforms strong baselines across multiple benchmarks, improving Pass@256 from 50.0% to 63.3% on AIME25 while effectively mitigating entropy collapse.

The main contributions of this work are as follows:

- We reveal the structural problem of binary reward signals losing semantic geometric information in RLVR, and discover that correctness labels and semantic distance constitute two orthogonal dimensions that jointly determine a response's exploration value. Based on this, we propose a semantic geometry-based four-category framework, providing a new perspective for understanding diversity collapse.

- A simple yet effective solution, SDAE: a semantic geometry-based reward shaping method. By computing responses' semantic gravity to the positive centroid, SDAE achieves differentiated advantage modulation across four categories, adaptively balancing exploration and exploitation without introducing additional hyperparameters.

- Extensive experiments validate SDAE's effectiveness: consistently outperforming strong baseline methods across multiple competition-level reasoning benchmarks, achieving up to 13.3% absolute improvement in Pass@256 on AIME25, demonstrating powerful reasoning scaling capabilities.

## 2. Related Work

### 2.1. Reinforcement Learning for LLM reasoning

Reinforcement learning (RL) has become a core mechanism for enhancing the reasoning and decision-making capabilities of LLMs, extending their abilities beyond supervised

fine-tuning (Christiano et al., 2017; Ouyang et al., 2022; Rafailov et al., 2023). Recently, RLVR has emerged as a promising paradigm for enhancing LLM reasoning capabilities, particularly in math and coding (Shao et al., 2024; Guo et al., 2025; Lambert et al., 2024). Unlike traditional RLHF, which relies on reward models trained on human preferences, RLVR employs deterministic verification functions to evaluate response correctness. This simple yet effective mechanism offers multiple advantages: mitigating reward hacking by providing ground-truth feedback, eliminating the need for extensive human annotation and complex reward model training, and demonstrating significant sample efficiency (Zhu et al., 2025). OpenAI o1 (Jaech et al., 2024) demonstrates that RL can effectively incentivize reasoning capabilities at scale, followed by models such as DeepSeek-R1 (Guo et al., 2025), QwQ (Team, 2025), Kimi k1.5 (Team et al., 2025), and Qwen3 (Yang et al., 2025).

### 2.2. Exploration Strategies in Reinforcement Learning

Exploration strategies in RLVR have evolved through multiple paradigms. Entropy-based methods prevent policy collapse by adding entropy rewards, but RLVR often leads to entropy collapse, where probability mass concentrates on limited tokens (Haarnoja et al., 2018; Yu et al., 2025; Li et al., 2025; Jin et al., 2025). Research shows that positive advantage tokens are the primary contributors, with most entropy consumption occurring in early training stages (Cui et al., 2025b; Wang et al., 2025a; Liang et al., 2025). Negative sample reinforcement (NSR) provides an alternative, where training solely on negative samples can improve Pass@$k$ by suppressing incorrect responses and redistributing probability mass to reasonable alternatives (Zhu et al., 2025; Tang et al., 2025). Semantic-level diversity methods have recently emerged, including EVOL-RL's novelty-aware rewards based on semantic dissimilarity (Zhou et al., 2025), DARLING's learned binary classifier for diversity measurement (Li et al., 2025), G²RL's first-order update geometry revealing external semantic embeddings as unreliable exploration proxies (Liang et al., 2025), and SEED-GRPO's semantic entropy regularization (Chen et al., 2025). Unlike these prior methods that rely on token-level entropy or polarity-based advantage shaping, we propose SDAE, which leverages geometric gravity continuous vector space for finer-grained exploration guidance.

## 3. Method

### 3.1. Motivation: The Semantic Geometry of Exploration

The core insight of our method is that model responses to the same question occupy different positions in semantic embedding space, with their geometric relationships encoding rich information about reasoning diversity. Standard RLVR methods apply uniform rewards to all correct responses and

uniform penalties to all incorrect responses, ignoring this semantic structure. However, we argue that this structure should be leveraged to guide exploration in RL.

We propose the centroid of correct responses $\mu^+$ as the gravitational source. Each response's distance to this gravitational source—termed Semantic Gravity ($SG$)—captures its degree of deviation from the "correctness center." This naturally produces distinctive topological signatures in the latent manifold, as visualized in Figure 1. We formalize this observation into a four-category system (Figure 2) that reveals the semantic geometry of exploration:

- **Trivial Correctness:** Correct responses clustered near the centroid represent conventional solution paths the model has already mastered. Reinforcing these is necessary, but over-rewarding risks diversity collapse.

- **Exploratory Correctness:** Correct responses far from the centroid represent novel, semantically diverse solution paths. The model has discovered a novel yet effective solution path; these should receive amplified rewards to encourage continued semantic diversification.

- **Local Error:** Incorrect responses still semantically close to correct solutions represent "near-misses." Strongly penalizing these would suppress promising reasoning directions; instead, weak penalties allow the model to refine these approaches.

- **Divergent Error:** Incorrect responses semantically distant from correct solutions represent fundamental reasoning failures. These should receive strong penalties to prevent the model from wasting capacity on ineffective reasoning directions.

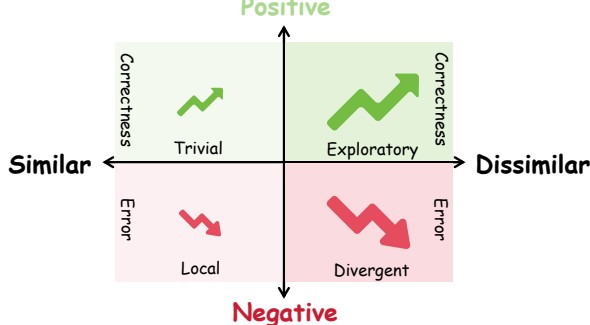

*Figure 2.* Classification of SDAE feedback. Arrow colors denote correctness (green) or error (red), while thickness represents reward/penalty magnitude.

This semantic geometry perspective enables SDAE to dynamically calibrate advantage values based on both correctness and semantic distance, balancing exploration and exploitation in a semantically meaningful way.

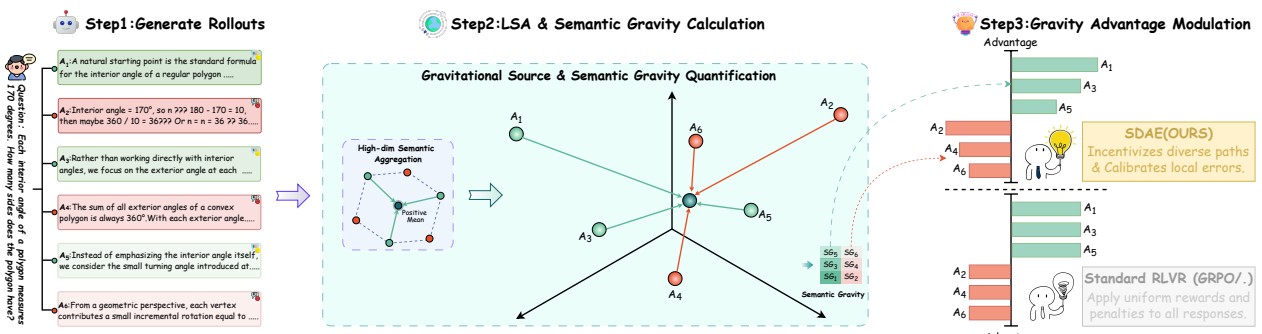

*Figure 3.* Illustration of the SDAE framework. The process encompasses LLM rollout generation, semantic gravity calculation, and advantage modulation. Unlike GRPO's uniform reward scheme, SDAE enables differentiated reinforcement guided by semantic geometry, encouraging diverse correct paths while suppressing divergent errors.

### 3.2. SDAE:Semantic-Diversity-Aware Exploration

The SDAE framework achieves dynamic reshaping of reinforcement learning advantage functions by perceiving the geometric topology of samples in latent space. Its core logic lies in using semantic gravity as a modulation factor to explicitly balance the model's "exploitation" on known high-frequency paths with "exploration" toward unknown semantic boundaries. As shown in Figure 3, the algorithm execution flow divides into three standardized phases:

**Phase 1: Rollout Sampling.** For a given question $q$, we first sample $N$ responses $\mathcal{Y} = \{y_1, \ldots, y_N\}$ from the current policy $\pi_\theta$.

**Phase 2: Latent Semantic Aggregation & Semantic Gravity Calculation.** To ensure that semantic representations remain aligned with the policy model's internal reasoning dynamics, we employ Latent Semantic Aggregation (LSA), which directly leverages the last-layer hidden states of the policy model. This design is theoretically motivated by recent findings demonstrating that the final layer serves as the unique optimization bottleneck and that external encoders exhibit geometric misalignment with the policy's gradient space (Liang et al., 2025). We provide a detailed discussion in Appendix A.

Let $\mathbf{h}_{i,t}$ denote the last-layer hidden state of the $i-th$ rollout at time step $t$. Its global semantic vector $v_i$ is defined as:

$$v_i = \text{MeanPool}(\{\mathbf{h}_{i,t}\}_{t=1}^{T_i}) \in \mathbb{R}^d. \quad (1)$$

Simultaneously, based on verifiable rewards $r_i \in \{+1, 0\}$, we partition samples into correct and incorrect sets:

$$\mathcal{Y}^+ = \{y_i : r_i = +1\}, \mathcal{Y}^- = \{y_i : r_i = 0\}. \quad (2)$$

Let $\mathcal{P} = \{y_i : r_i = 1\}$ denote the set of correct responses. The gravitational source is:

$$\boldsymbol{\mu}^+ = \frac{1}{|\mathcal{P}|} \sum_{y_i \in \mathcal{P}} v_i. \quad (3)$$

Subsequently, we compute the semantic gravity for each sample $y_i$, i.e., its degree of deviation from the consensus path on the cosine manifold:

$$SG_i = 1 - \frac{v_i \cdot \mu^+}{\|v_i\|\|\mu^+\|}. \quad (4)$$

Smaller $SG_i$ indicates the response is semantically closer to the centroid, while larger $SG_i$ indicates semantic deviation.

**Phase 3: Four-Category Advantage Modulation.** Unlike the uniform rewards applied by standard GRPO (Shao et al., 2024), SDAE introduces the modulation coefficient $SG$ to reweight the original advantage function $A_i$ (see Appendix E for the complete GRPO formulation):

$$A_i^{SDAE} = A_i \cdot SG_i. \quad (5)$$

The SDAE-modulated GRPO objective then becomes:

$$\mathcal{J}_{SDAE}(\theta) = \mathbb{E}_{[q \sim P(Q), \{o_i\}_{i=1}^G \sim \pi_{\theta_{old}}(O|q)]} \frac{1}{G} \sum_{i=1}^G \frac{1}{|o_i|} \sum_{t=1}^{|o_i|} \left\{ \min\left( \frac{\pi_\theta(o_{i,t}|q, o_{i,<t})}{\pi_{\theta_{old}}(o_{i,t}|q, o_{i,<t})} \hat{A}_{i,t}^{SDAE}, \right. \right.$$
$$\left. \left. \text{clip}\left( \frac{\pi_\theta(o_{i,t}|q, o_{i,<t})}{\pi_{\theta_{old}}(o_{i,t}|q, o_{i,<t})}, 1 - \varepsilon, 1 + \varepsilon \right) \hat{A}_{i,t}^{SDAE} \right) - \beta \mathbb{D}_{KL}[\pi_\theta||\pi_{ref}] \right\}. \quad (6)$$

This simple linear reweighting mechanism mathematically and automatically implements the four-category exploration logic we propose. See Appendices B-D for further implementation details.

### 3.3. Theoretical Justification of SDAE

To delve deeper into SDAE's internal mechanism, we analyze how semantic gravity modulates policy gradient updates to elucidate its logic for balancing exploration and

exploitation in semantic manifolds. For clarity, consider the loss function without clipping:

$$L_i(\theta) = \text{ratio}_i(\theta) \cdot \hat{A}_i = \frac{\pi_\theta(o_i|q)}{\pi_{\theta_{\text{old}}}(o_i|q)} \cdot \hat{A}_i. \qquad (7)$$

The gradient is computed as:

$$\nabla_\theta L_i(\theta) = \nabla_\theta \log \pi_\theta(o_i|q) \cdot \text{ratio}_i(\theta) \cdot \hat{A}_i. \qquad (8)$$

Correspondingly, the policy update becomes (with global learning rate $\eta$):

$$\theta \leftarrow \theta + \eta \cdot \nabla_\theta \log \pi_\theta(o_i|q) \cdot \text{ratio}_i(\theta) \cdot \underbrace{A_i \cdot \text{SG}_i}_{\hat{A}_i}. \qquad (9)$$

As shown above, policy updates are controlled by four components: the global learning rate $\eta$, the log-probability gradient $\nabla_\theta \log \pi_\theta(o_i|q)$, the importance sampling ratio $\text{ratio}_i(\theta)$, and the semantically-aware advantage $\hat{A}_i = A_i \cdot SG_i$. By integrating the semantic gravity factor, SDAE effectively modulates update magnitudes according to each response's geometric position in semantic space. This can be viewed as dynamically adjusting the effective learning rate at the response level: the original advantage $A_i$ determines the direction of parameter updates (reinforcement or suppression), while $SG_i$ scales the magnitude according to semantic distance from the positive centroid.

This design produces an important property: responses semantically close to known correct solutions receive attenuated gradients regardless of their correctness labels. For correct responses, this avoids redundant reinforcement of already well-captured patterns. For incorrect responses, this preserves exploration potential in promising semantic regions, as minor adjustments may lead to correct solutions. Conversely, responses far from the positive centroid receive amplified gradients, enabling the model to strongly reinforce novel correct reasoning while decisively suppressing fundamentally flawed paths. This adaptive modulation creates an implicit curriculum where the model efficiently learns from semantically informative samples while avoiding overfitting to redundant or noisy signals.

## 4. Experiments

### 4.1. Experimental Setup

**Training Datasets.** To validate SDAE's robustness across different training settings, we conduct experiments on two representative datasets. **MATH** (Hendrycks et al., 2021) contains 7,500 complex competition mathematics problems

spanning algebra, geometry, number theory, and other domains. Following prior work (Zhu et al., 2025), we use the training set for reinforcement learning. **DAPO-Math-17K** (Yu et al., 2025) is a dataset containing 17,000 mathematics problems with integer answers, specifically designed for large-scale LLM reinforcement learning and carefully curated to ensure accurate reward signals. In accordance with Cheng et al. (2025), we adopt the same prompt templates and verifiers for experiments.

**Model and Baselines.** Following prior work (Zhu et al., 2025), we use Qwen2.5-Math-7B (Yang et al., 2024) as our base model. We compare SDAE against mainstream RL algorithms and recent exploration-enhanced methods, including GRPO, Entropy Adv, PSR, NSR, and W-REINFORCE. PSR and NSR selectively update the policy model using only correct or incorrect responses, respectively. Entropy Adv encourages longer reasoning chains by adding token-level entropy rewards $\psi(H)$. W-REINFORCE is a REINFORCE variant that reduces the weight of the positive sample reinforcement component while maintaining negative sample reinforcement, balancing accuracy and diversity. More details are provided in Appendix E.

**RL Training Configuration.** We train models using the VERL framework (Sheng et al., 2025). For both datasets, we use the prompt templates shown in Appendix F and implementation details in Appendix G to further elaborate the training setup.

**Evaluation Benchmarks and Metrics.** We evaluate on AIME 2024/2025 (MAA, 2025), AMC23 (MAA, 2023), and MATH/MATH500 (Hendrycks et al., 2021), using a sampling temperature of 0.6, maximum response length of 4K tokens, and top-p sampling with p=0.95. Following Zhu et al. (2025), we use the Pass@$k$ metric to evaluate reasoning capability boundaries, with details in Appendix I.

### 4.2. Main Results

As shown in Table 1, we report Pass@$k$ and Pass@1 results for Qwen2.5-Math-7B trained on DAPO-Math-17K. GRPO w/ SDAE consistently outperforms baseline methods across all benchmarks. On AIME 2025, SDAE achieves Pass@256 of 63.3% and Pass@1 of 20.0%, representing improvements of +5.9% and +3.7% over vanilla GRPO, respectively. Compared to GRPO w/ Entropy Advantage, SDAE's Pass@1 is +2.4% higher, demonstrating that semantic-level diversity provides more effective exploration signals than token-level entropy. Notably, on AMC23, SDAE achieves a +12.8% Pass@1 improvement, substantially surpassing entropy-based methods (+2.9%). SDAE also matches or exceeds multiple strong baselines across benchmarks, including Qwen2.5-Math-Instruct, Eurus-2-PRIME, Oat-Zero, and GPG, validating its effectiveness as a general exploration enhancement technique for LLM reinforcement learning.

*Table 1.* **Pass@k and Pass@1 results on MATH-500, AIME 2024/2025 and AMC23.** †: results from Chu et al. (2025), ‡: results from Cheng et al. (2025). **Green** type denotes the performance gain of SDAE (blue background) over related baselines (gray background).

| Method | Venue | AIME25 | | AIME24 | | AMC23 | | MATH500 | |
|--------|-------|---------|---------|---------|---------|---------|---------|---------|---------|
| | | Pass@256 | Pass@1 | Pass@256 | Pass@1 | Pass@128 | Pass@1 | Pass@16 | Pass@1 |
| Qwen2.5-Math | Qwen | 53.3 | 6.2 | 70.0 | 14.2 | 90.0 | 40.3 | 90.6 | 57.2 |
| Qwen2.5-Math-Ins | Qwen | 53.3 | 8.7 | 60.0 | 11.3 | 92.4 | 54.7 | 93.5 | 79.8 |
| Eurus-2-PRIME† | arXiv'25 | - | - | - | 26.7 | - | 57.8 | - | 79.2 |
| Oat-Zero† | COLM'25 | - | - | - | 30.0 | - | 55.4 | - | 80.6 |
| GPG† | arXiv'25 | - | - | - | 33.3 | - | 65.0 | - | 80.0 |
| + GRPO‡ | | 57.4 | 16.3 | 83.3 | 30.9 | 92.8 | 66.9 | 94.6 | 83.0 |
| + GRPO w/ Entropy Adv‡ | AAAI'26 | 63.6 | 17.6 | 80.0 | 33.7 | 95.2 | 69.8 | 94.8 | 83.1 |
| **+ GRPO w/ SDAE** | **Ours** | **63.3** | **20.0** | **80.0** | **32.9** | **95.6** | **79.7** | **95.4** | **85.2** |
| *△ Gain* | | **+5.9** | **+3.7** | **-3.3** | **+2.0** | **+2.8** | **+12.8** | **+0.8** | **+2.2** |

*Table 2.* **Pass@k results on MATH, AIME 2025 and AMC23 with Qwen2.5-Math-7B.** **Bold** and underlined types denote the best and second-best results for each k. ⋆: results from Zhu et al. (2025).

| Method | Pass@k | | | | | | | | |
|--------|--------|---|---|---|---|---|---|---|---|
| k | 1 | 2 | 4 | 8 | 16 | 32 | 64 | 128 | 256 |
| *MATH* | | | | | | | | | |
| Base Model⋆ | 63.2 | 76.0 | 83.7 | 88.4 | 91.6 | 93.7 | 95.2 | 96.2 | 96.9 |
| GRPO⋆ | 76.3 | 81.7 | 85.6 | 88.4 | 90.6 | 92.3 | 93.6 | 94.7 | 95.5 |
| PPO⋆ | 76.6 | 82.6 | 86.7 | 89.6 | 91.7 | 93.4 | 94.7 | 95.6 | 96.3 |
| PSR⋆ | 74.1 | 78.3 | 81.6 | 84.1 | 86.2 | 87.9 | 89.3 | 90.4 | 91.2 |
| NSR⋆ | 75.7 | 82.4 | 86.9 | 90.1 | 92.4 | 94.1 | 95.3 | 96.2 | 96.9 |
| W-REINFORCE⋆ | 76.6 | 82.8 | 87.1 | 90.2 | 92.4 | 94.1 | 95.3 | 96.1 | 96.7 |
| REINFORCE w/ SDAE | **81.0** | **85.0** | **89.3** | **93.2** | **97.8** | **98.3** | **98.7** | **98.9** | **99.2** |
| GRPO w/ SDAE | 77.1 | 83.5 | 87.1 | 90.7 | 92.1 | 95.0 | 95.9 | 96.6 | 97.2 |
| *AIME 2025* | | | | | | | | | |
| Base Model⋆ | 6.1 | 9.7 | 13.8 | 17.9 | 22.2 | 26.5 | 30.8 | 36.6 | 46.7 |
| GRPO⋆ | 10.3 | 14.7 | 19.4 | 24.0 | 28.4 | 32.8 | 37.3 | 42.5 | 50.0 |
| PPO⋆ | 8.5 | 13.2 | 18.0 | 22.5 | 26.6 | 30.3 | 33.8 | 37.9 | 43.3 |
| PSR⋆ | 11.6 | 14.1 | 16.2 | 18.6 | 21.7 | 25.7 | 30.9 | 36.9 | 43.3 |
| NSR⋆ | 10.0 | 14.6 | 19.2 | 24.1 | 29.3 | 34.6 | 40.2 | 46.0 | 53.3 |
| W-REINFORCE⋆ | 10.6 | 15.3 | 20.0 | 24.7 | 29.7 | 34.6 | 40.5 | 47.8 | 56.7 |
| REINFORCE w/ SDAE | **13.2** | **18.0** | **22.4** | **26.9** | **31.9** | **37.9** | **45.2** | **54.6** | **66.7** |
| GRPO w/ SDAE | 11.7 | 15.8 | 20.0 | 24.5 | 30.0 | 36.3 | 44.1 | 53.6 | 63.3 |
| *AMC23* | | | | | | | | | |
| Base Model⋆ | 41.0 | 56.2 | 69.2 | 78.9 | 85.1 | 89.1 | 92.9 | 97.2 | **100.0** |
| GRPO⋆ | 61.7 | 68.7 | 74.6 | 80.0 | 85.1 | 89.7 | 93.4 | 95.9 | 97.5 |
| PPO⋆ | 62.0 | 70.0 | 76.1 | 80.9 | 85.3 | 89.5 | 93.1 | 96.0 | 97.5 |
| PSR⋆ | **62.6** | 69.9 | 74.5 | 77.5 | 80.3 | 83.5 | 87.2 | 90.6 | 92.5 |
| NSR⋆ | 60.9 | 70.0 | 77.4 | 83.2 | 87.6 | 91.1 | 94.5 | 97.9 | 100.0 |
| W-REINFORCE⋆ | 62.0 | 70.0 | 77.0 | 83.1 | 87.8 | 91.8 | 95.2 | 97.1 | 97.5 |
| REINFORCE w/ SDAE | 61.4 | 70.1 | 77.0 | 82.8 | 87.6 | 92.2 | **96.4** | **99.3** | **100.0** |
| GRPO w/ SDAE | 62.5 | **70.6** | **77.5** | **83.5** | **88.3** | **92.6** | 96.1 | 98.9 | **100.0** |

Table 2 presents Pass@k results trained on full MATH dataset. REINFORCE w/ SDAE achieves the best results across nearly all k values on MATH (Pass@1 81.0%, Pass@256 99.2%) and AIME 2025 (Pass@1 13.2%, Pass@256 66.7%). Both SDAE variants achieve 100.0% Pass@256 on AMC23, while multiple baselines fail to maintain this exploration capability. Unlike PSR, which improves Pass@1 but degrades high-k performance, SDAE improves across the entire Pass@k spectrum. On AIME 2025, SDAE not only surpasses RL baselines but also breaks through the base model's Pass@256 ceiling, demonstrating its ability to transcend the base model's inherent reasoning boundaries.

### 4.3. Analysis of Exploration Capability via Pass@k

To better understand how SDAE affects model exploration capability, we analyze Pass@k curves across different k values. As shown in Figure 4 (on DAPO-Math-17K) and Table 2 (on MATH), we compare the inference-time scaling behavior of Base Model, GRPO, and GRPO w/ SDAE.

**SDAE improves performance across the entire Pass@k spectrum.** As shown in Figure 4, on AIME 2025, GRPO w/ SDAE consistently outperforms both the base model and vanilla GRPO across all k values (16 to 256). At k=256, SDAE achieves 63.3%, surpassing GRPO (57.4%) and the base model (53.3%). Similar trends are observed on AIME 2024, AMC23, and MATH500. This indicates that SDAE not only improves Pass@1 accuracy but also elevates the model's reasoning ceiling under more attempts.

**SDAE preserves exploration capability lost by standard RL.** A key finding from Table 2 is that standard RL methods often achieve lower Pass@k at high k values than the base model. On MATH, the base model achieves 96.9%

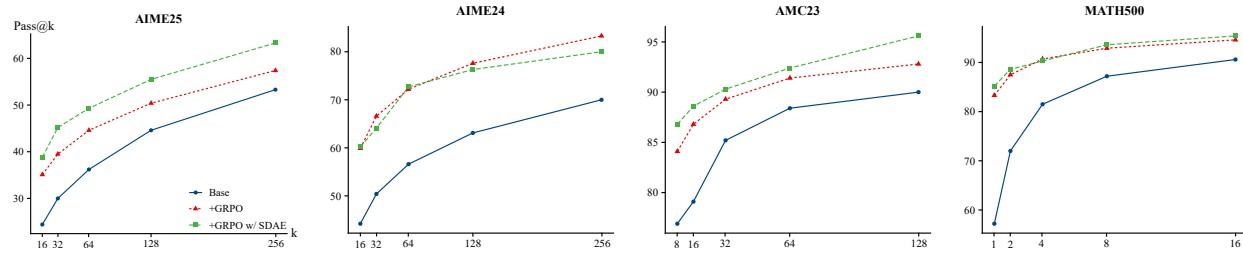

*Figure 4.* **Pass@$k$ performance of Qwen2.5-Math-7B on DAPO-Math-17K** using different RL algorithms across various benchmarks.

Pass@256, while GRPO drops to 95.5% and PSR drops to 91.2%. In contrast, SDAE variants maintain or exceed the base model's ceiling: REINFORCE w/ SDAE achieves 99.2%, GRPO w/ SDAE achieves 97.2%. On AMC23, both SDAE variants achieve perfect 100.0% Pass@256, while PPO (97.5%) and PSR (92.5%) fail to preserve the base model's full exploration capability.

**Semantic diversity achieves better exploration-exploitation balance.** The Pass@$k$ curves reveal that SDAE achieves a superior balance: it improves low-$k$ performance (exploitation) without sacrificing high-$k$ performance (exploration). This contrasts with PSR, which aggressively improves Pass@1 but collapses diversity at high $k$. By weighting advantages based on semantic gravity rather than token-level entropy, SDAE encourages diverse solution paths while still reinforcing correct solutions.

### 4.4. Entropy Dynamics: Mitigating Strategy Collapse

To further investigate how SDAE prevents models from falling into local optima, we monitor policy entropy throughout training on the DAPO-MATH-17K dataset. Policy entropy is a core metric for measuring model exploration potential. As shown in Figure 5, we compare SDAE against vanilla GRPO and state-of-the-art entropy control methods Clip-Cov and KL-Cov (Cui et al., 2025b).

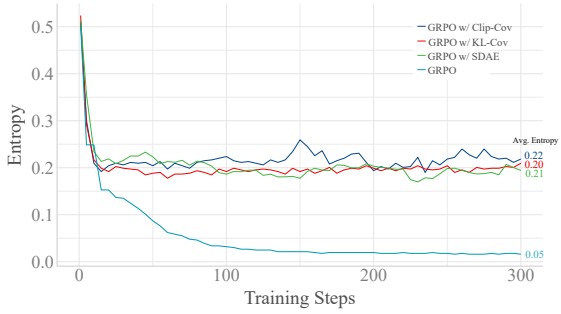

*Figure 5.* Policy entropy dynamics on the DAPO-MATH-17K dataset. SDAE maintains a stable entropy level (0.21) throughout training, effectively mitigating the catastrophic entropy collapse seen in vanilla GRPO (0.05). Its performance is comparable to gradient-level regularization methods like Clip-Cov and KL-Cov.

In vanilla GRPO (light blue curve), we observe sharp and monotonic entropy decline, collapsing to approximately 0.05 within 300 training steps. This "entropy collapse" phenomenon indicates the policy has become overconfident on a very limited set of solution paths, leading to loss of exploration capability and performance saturation.

In contrast, our proposed SDAE (green curve) effectively mitigates this collapse trend, stabilizing average entropy around 0.21. Unlike token-level interventions, SDAE operates at the response level. By considering global semantic structure, SDAE preserves diversity among semantically distinct solutions.

Compared to methods like Clip-Cov (0.22) and KL-Cov (0.20) that explicitly constrain high-covariance token updates, SDAE achieves comparable entropy maintenance without requiring complex per-token covariance computations. This demonstrates that encouraging semantic-level diversity can provide a global exploration signal that is equally effective as specialized gradient-level constraints in maintaining model "vitality" and preventing policy collapse.

### 4.5. Analysis of Training Dynamics and Behavior Evolution

To deeply investigate SDAE's learning mechanism, we track four metrics during training on the MATH dataset: (1) greedy decoding accuracy on the test set, (2) policy entropy, (3) the ratio of correct responses per batch on the training set, and (4) the proportion of fully-solved prompts per batch (i.e., all rollouts are correct) on the training set. These metrics reveal how different objectives shape the balance between exploration and exploitation, with results shown in Figure 6.

As shown in Figure 6a, SDAE-enhanced models achieve significantly better greedy decoding accuracy than vanilla GRPO and NSR. Although PSR shows faster accuracy growth in early training, it quickly plateaus due to falling into the "pure exploitation" trap (i.e., overfitting to initial correct paths at the expense of generalized reasoning capability). In contrast, SDAE maintains steady upward trends throughout training, demonstrating superior convergence.

Figure 6b shows model exploration potential through pol-

icy entropy. Both vanilla GRPO and PSR exhibit severe entropy collapse (dropping below 0.05), indicating loss of generation diversity. While NSR maintains higher entropy by suppressing incorrect samples, its accuracy improvement is slower. SDAE successfully achieves the optimal balance: it maintains significantly higher and stable entropy levels than vanilla GRPO (0.15), preserving sufficient reasoning boundaries while optimizing accuracy, avoiding the policy collapse common in standard RLVR.

The trends in Figure 6c and d indicate that SDAE is more efficient than NSR at discovering correct solution paths. By using semantic gravity to guide exploration, SDAE is more aggressive than pure negative sample reinforcement at improving correct sample ratios and fully-solved ratios. This confirms that rewarding semantic-level diversity is a more potent signal for eliciting latent model reasoning behaviors compared to mere failure penalties.

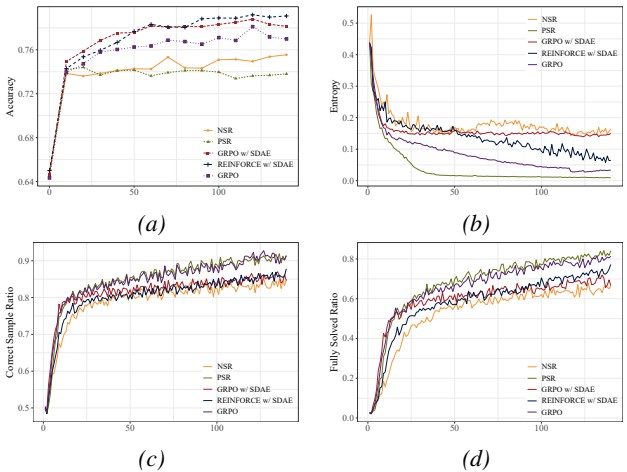

*(a)*      *(b)*

*(c)*      *(d)*

*Figure 6.* Training dynamics on MATH dataset under GRPO, PSR, NSR, GRPO w/ SDAE and REINFORCE w/ SDAE across training steps, including (a) greedy decoding accuracy on the math test set, (b) the model's entropy, (c) the ratio of correct responses per batch on the training set, and (d) the proportion of fully-solved prompts per batch (i.e., all rollouts are correct) on the training set.

### 4.6. Impact of Embedding Spaces on Semantic Exploration

To validate SDAE's robustness and the impact of different embedding spaces on semantic exploration, we compare the default Latent Semantic Aggregation (LSA) against External Semantic Encoders (ESE) using external embedding models of varying scales (Qwen3-Embedding-0.6B and 4B).

As shown in Table 3, the LSA method consistently attains the best or highly competitive performance across both benchmarks. Specifically, on AIME25, LSA achieves 63.3% Pass@256, outperforming the 4B ESE version (60.0%) and 0.6B ESE version (56.7%). This indicates that the policy

model's own latent space already adequately models reasoning diversity, and leveraging internal states mitigates potential distribution shifts that may be introduced by misaligned external encoders.

Among ESE variants, we observe clear scaling laws: scaling the external encoder parameters from 0.6B to 4B yields monotonic improvements in Pass@$k$ results. For example, on AMC23, the 4B ESE version achieves 96.0% Pass@64, significantly higher than the 0.6B version's 93.9%.

Despite different embedding sources, all SDAE configurations significantly outperform the vanilla GRPO baseline at higher $k$ values. LSA's superior performance demonstrates that the "semantic geometry" required for effective exploration is already intrinsically captured in the policy's own reasoning process, making SDAE an efficient and self-contained framework that does not strictly depend on additional heavy external models.

*Table 3.* Performance comparison (Pass@$k$) on AIME25 and AMC23 benchmarks. We compare our SDAE framework using LSA against ESE with varying model scales. The red ✗ indicates that no external model is used. Light blue and light yellow indicate the best performance in each column.

| Method | Settings | Pass@$k$ | | | | | | | | |
|---|---|---|---|---|---|---|---|---|---|---|
| | | 1 | 2 | 4 | 8 | 16 | 32 | 64 | 128 | 256 |
| **AIME25** | | | | | | | | | | |
| Base Model | ✗ | 6.1 | 9.7 | 13.8 | 17.9 | 22.2 | 26.5 | 30.8 | 36.6 | 46.7 |
| GRPO | ✗ | 10.3 | 14.7 | 19.4 | 24.0 | 28.4 | 32.8 | 37.3 | 42.5 | 50.0 |
| GRPO w/ SDAE (LSA) | ✗ | 11.1 | 15.5 | 20.0 | 24.5 | 30.0 | 36.3 | 44.1 | 53.6 | 63.3 |
| GRPO w/ SDAE (ESE) | Qwen3-Embedding-0.6B | 9.6 | 13.4 | 18.1 | 23.2 | 28.1 | 33.9 | 40.9 | 48.7 | 56.7 |
| | Qwen3-Embedding-4B | 10.6 | 14.2 | 19.0 | 24.0 | 29.7 | 36.3 | 43.1 | 50.4 | 60.0 |
| **AMC23** | | | | | | | | | | |
| Base Model | ✗ | 41.0 | 56.2 | 69.2 | 78.9 | 85.1 | 89.1 | 92.9 | 97.2 | 100.0 |
| GRPO | ✗ | 61.7 | 68.7 | 74.6 | 80.0 | 85.1 | 89.7 | 93.4 | 95.9 | 97.5 |
| GRPO w/ SDAE (LSA) | ✗ | 61.1 | 70.2 | 77.4 | 83.5 | 87.8 | 92.0 | 95.9 | 98.9 | 100.0 |
| GRPO w/ SDAE (ESE) | Qwen3-Embedding-0.6B | 60.9 | 69.7 | 76.7 | 81.8 | 86.1 | 90.2 | 93.9 | 96.5 | 97.5 |
| | Qwen3-Embedding-4B | 61.9 | 70.2 | 76.6 | 81.9 | 86.0 | 90.8 | 96.0 | 99.3 | 100.0 |

## 5. Conclusion

This paper proposes Semantic-Diversity-Aware Exploration (SDAE), a novel method that guides LLM reasoning exploration through semantic space geometry. Our core insight is that binary reward signals discard the semantic geometric information of the response space, while correctness and semantic distance constitute two orthogonal dimensions that jointly determine a response's exploration value. SDAE differentially modulates advantage functions by computing responses' semantic distance to the positive centroid, adaptively balancing exploration and exploitation without additional hyperparameters. Experiments demonstrate that SDAE outperforms strong baselines across multiple benchmarks, achieving a 13.3% improvement in Pass@256 on AIME25, effectively breaking through the reasoning ceiling of the baseline model. Our work demonstrates that the geometric structure of semantic space can precisely capture the nuanced exploration value of responses, providing a new perspective for balancing exploration and exploitation.

## Impact Statement

This paper presents work whose goal is to advance the field of Machine Learning. There are many potential societal consequences of our work, none which we feel must be specifically highlighted here.

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

## A. Theoretical Connection to Gradient-Guided Exploration

We provide additional context on the theoretical relationship between our Latent Semantic Aggregation (LSA) approach and recent work on gradient-guided exploration.

$G^2$RL (Liang et al., 2025) analyzes the source of exploration signals from the perspective of first-order update geometry, demonstrating that external semantic embeddings are inherently flawed as proxies for exploration. This provides theoretical justification for using the policy model's internal representations. Specifically, G²RL proves that for autoregressive language models, parameter gradients at any layer can be decomposed as:

$$\nabla_{\theta_k} \ell(x, y) = \frac{1}{T} \mathcal{L}_k(x, y) \Phi(x, y), \tag{10}$$

where $\Phi(x, y) = \sum_t W(e(y_t) - p_t)$ is a sequence-level gradient sensitivity feature derived from the last-layer, and $\mathcal{L}_k(x, y)$ is a trajectory-dependent linear operator. This factorization theorem reveals that the last-layer serves as the unique first-order sensitivity bottleneck through which all trajectory-specific optimization information must pass.

It is worth noting that while G²RL's sequence feature $\Phi(x, y)$ represents an exact gradient sensitivity measure, our LSA employs a simplified approximation using mean-pooled hidden states: $v_i = \text{MeanPool}(\{h_{i,t}\}_{t=1}^{T_i})$. Despite this difference in mathematical formulation, the core insight remains consistent: the policy model's internal representation space reflects optimization-relevant geometric structure more faithfully than external encoders. Our ablation study in Table 3 empirically validates the effectiveness of this design choice.

Furthermore, while G²RL guides exploration directly in the gradient feature space, our SDAE operates in the semantic embedding space, achieving finer-grained advantage modulation through the gravitational mechanism. This distinction allows SDAE to focus on the semantic diversity of reasoning paths rather than raw gradient orthogonality, which we find to be more effective for incentivizing novel solution strategies while suppressing divergent errors.

## B. Handling Homogeneous Rollouts (All-Correct or Zero-Correct Cases)

In the practical implementation of SDAE, we encounter cases where all $N$ rollouts for a single prompt $q$ are either entirely correct ($r_i = 1, \forall i$) or entirely incorrect ($r_i = 0, \forall i$). We handle these homogeneous groups differently depending on the underlying RL algorithm:

### B.1. Integration with GRPO

For GRPO w/ SDAE, we strictly follow the advantage computation logic of vanilla GRPO. The standard advantage $A_i$ is defined by normalizing the rewards within the sampled group:

$$A_i = \frac{R(o_i) - \text{mean}(\{R(o)\})}{\text{std}(\{R(o)\})}. \tag{11}$$

When all responses share the same reward label (e.g., all correct or all incorrect), the standard deviation becomes zero (or the numerator vanishes), resulting in $A_i = 0$ for all $i$. Consequently, the semantically-aware advantage also becomes zero:

$$A_i^{SDAE} = A_i \cdot SG_i = 0. \tag{12}$$

As a result, these samples provide no gradient signal and do not trigger policy updates. This design ensures that SDAE maintains consistency with the baseline GRPO mechanism without introducing unintended optimization biases.

### B.2. Integration with REINFORCE

In REINFORCE w/ SDAE, the calculation of the gravitational source $\mu^+$ typically requires the set of correct responses $\mathcal{Y}^+$. We handle the edge cases as follows:

- **All-Correct Case:** If all rollouts are correct ($\mathcal{Y}^+ = \mathcal{Y}$), the centroid $\mu^+$ is calculated as the mean-pool of all $N$ sampled hidden states $v_i$.

- **Zero-Correct (All-Wrong) Case:** When no correct response is found ($\mathcal{Y}^+ = \emptyset$), the standard definition of $\mu^+$ becomes undefined. In this scenario, we adopt a consensus-based fallback, using the centroid of all sampled rollouts as the proxy gravitational source:

$$\mu^+ = \frac{1}{N} \sum_{i=1}^{N} v_i \quad \text{if} \quad \mathcal{Y}^+ = \emptyset. \tag{13}$$

This allows the model to still distinguish between Local Errors (semantically near the group's consensus) and Divergent Errors. By attenuating penalties for local errors relative to the group consensus, we preserve the model's potential for iterative refinement even when no correct path is discovered in the current batch.

## C. Normalization of Semantic Gravity

In our implementation, the Semantic Gravity ($SG$) is not mapped to a fixed range via hard clipping. Instead, we employ a batch-level mean normalization strategy to modulate the advantages. Specifically, for a set of sampled responses for a prompt, the raw semantic distances are scaled by their batch mean:

$$\hat{SG}_i = \frac{SG_i}{\frac{1}{N} \sum_{j=1}^{N} SG_j + \epsilon}, \tag{14}$$

where $SG_i$ represents the raw cosine distance between the response embedding and the positive centroid. This normalization provides several key benefits:

- **Scale Invariance:** By ensuring the average multiplier is approximately $1.0$, we preserve the overall magnitude of the gradients, preventing the advantage values from vanishing or exploding as the model's embedding space shifts.

- **Relative Novelty:** This approach emphasizes the relative semantic novelty of a response within its current sampling group rather than its absolute position in the embedding space.

- **Stability:** It avoids the information loss associated with hard clipping (e.g., mapping all distant samples to $1.0$), allowing the model to distinguish between "moderately novel" and "highly divergent" paths.

## D. Analysis of Computational and Memory Overhead

The overhead introduced by Latent Semantic Aggregation (LSA) is negligible compared to the standard training and inference costs of LLMs. We analyze this from two perspectives:

- **Computational Efficiency:**
  - **Intrinsic Reuse:** LSA directly utilizes the last-layer hidden states already generated during the policy model's forward pass. Since these states are required for generating logits, LSA incurs zero additional inference cost.
  - **Low Complexity:** The mean pooling operation $v_i = \text{MeanPool}(\{h_{i,t}\})$ has a linear time complexity of $O(T \times d)$ (where $T$ is sequence length and $d$ is hidden dimension). This is several orders of magnitude smaller than the $O(T^2 \times d)$ complexity of the self-attention mechanism.
  - **Self-Contained Advantage:** Unlike methods requiring External Semantic Encoders (ESE), SDAE avoids the massive computational burden of running an additional 0.6B or 4B parameter model for every rollout.

- **Memory Efficiency:**
  - **Vector Compression:** While storing per-token hidden states for a full batch might seem demanding, SDAE only requires the pooled semantic vector $v_i \in \mathbb{R}^d$ for advantage modulation.
  - **Immediate Release:** The high-dimensional per-token tensors can be discarded immediately after the mean-pooling step during the rollout phase. The resulting storage overhead is merely $N \times d$ per prompt (where $N$ is the number of rollouts), which is marginal compared to the memory occupied by model weights and KV caches.

# E. Detailed Description of Baselines

In this section, we provide detailed descriptions of baseline methods used for comparison in our experiments. Specifically, we compare our method against GRPO (Shao et al., 2024), PRIME (Cui et al., 2025a), Oat-Zero (Liu et al., 2025), GPG (Chu et al., 2025), polarity-level advantage shaping methods (i.e., PSR, NSR, and W-REINFORCE (Zhu et al., 2025)), and token-level advantage shaping methods (i.e., w/ Entropy Advantage (Cheng et al., 2025)).

- **GRPO** (Shao et al., 2024) was proposed in DeepSeekMath as a simplified alternative to PPO, eliminating the need for a separate critic model. Instead of learning a value function, GRPO uses the mean reward within a sampled group as the baseline.

  **Objective function:**

  The GRPO objective maximizes the following clipped surrogate objective:

  $$\mathcal{J}_{GRPO}(\theta) = \mathbb{E}_{[q \sim P(Q), \{o_i\}_{i=1}^{G} \sim \pi_{\theta_{old}}(O|q)]}$$
  $$\frac{1}{G} \sum_{i=1}^{G} \frac{1}{|o_i|} \sum_{t=1}^{|o_i|} \left\{ \min \left( \frac{\pi_\theta(o_{i,t}|q, o_{i,<t})}{\pi_{\theta_{old}}(o_{i,t}|q, o_{i,<t})} \hat{A}_{i,t}, \text{clip} \left( \frac{\pi_\theta(o_{i,t}|q, o_{i,<t})}{\pi_{\theta_{old}}(o_{i,t}|q, o_{i,<t})}, 1 - \epsilon, 1 + \epsilon \right) \hat{A}_{i,t} \right) - \beta \mathbb{D}_{KL}[\pi_\theta || \pi_{ref}] \right\}, \tag{15}$$

  here, $\varepsilon$ is the clipping hyperparameter, $\beta$ is the KL penalty coefficient, and $\pi_{ref}$ denotes the reference policy.

  **Advantage computation:**

  Unlike PPO which learns a value function, GRPO computes advantages using group-level statistics:

  $$\hat{A}_i = \frac{R(o_i) - \text{mean}(\{R(o_1), \ldots, R(o_G)\})}{\text{std}(\{R(o_1), \ldots, R(o_G)\})}, \tag{16}$$

  where $G$ outputs are sampled for each question and scored by a reward model.

- **PRIME** (Cui et al., 2025a) introduces implicit process rewards, providing dense token-level feedback during RL training:

  **Core mechanism:**

  The implicit PRM computes process rewards as:

  $$r_\phi(y_t) = v_\phi(y_{<t+1}) - v_\phi(y_{<t}), \tag{17}$$

  where $v_\phi(y_{<t}) = \sum_{i=1}^{t-1} \beta \log \frac{\pi_\phi(y_i|y_{<i})}{\pi_{ref}(y_i|y_{<i})}$ is the implicit value function.

- **Oat-Zero** (Liu et al., 2025) demonstrates that RL can be directly applied to base models without SFT warm-up, while critically examining optimization biases in GRPO.

- **GPG** (Chu et al., 2025) proposes a simplified RL approach that directly optimizes the original objective without using surrogate losses:

  **Loss function:**

  $$\mathcal{L}_{\text{GPG}} = -\log \pi_\theta(o) \cdot A, \tag{18}$$

  where $A = \alpha \cdot (R(o) - \text{mean}(\{R(o)\}))$

- **PSR** and **NSR** are polarity-level advantage shaping methods proposed by Zhu et al. (2025), selectively updating policies based on response correctness.

- **W-REINFORCE** (Zhu et al., 2025) is a REINFORCE variant that reduces the weight of the positive sample reinforcement component while maintaining negative sample reinforcement, balancing accuracy and diversity.

  **Formulation:**

  The method reduces weights on positive advantage terms while maintaining full weights on negative advantages:

  $$\mathcal{L}_{\text{W-REINFORCE}} = -\mathbb{E}\left[w \cdot \mathbb{1}[A > 0] \cdot A \cdot \log \pi_\theta(o) + \mathbb{1}[A < 0] \cdot A \cdot \log \pi_\theta(o)\right], \tag{19}$$

  where $w < 1$ is the weight factor for positive samples.

- **Entropy Advantage** (Cheng et al., 2025) proposes encouraging exploratory reasoning behaviors by adding entropy-based terms to standard RL advantages.

  **Entropy-based advantage shaping:**

  For each token $o_t$, the entropy of the current policy over vocabulary $\mathcal{V}$ is:

  $$\mathcal{H}_t = -\sum_{v \in \mathcal{V}} \pi_\theta(v|q, o_{<t}) \log \pi_\theta(v|q, o_{<t}). \tag{20}$$

  The shaped advantage is:

  $$A_t^{\text{shaped}} = A_t + \psi(\mathcal{H}_t), \tag{21}$$

  where:

  $$\psi(\mathcal{H}_t) = \min\left(\alpha \cdot \mathcal{H}_t^{\text{detach}}, \frac{|A_t|}{\kappa}\right), \quad \alpha > 0, \kappa > 1$$

## F. Prompt Templates and Reward Function

We adopt the prompt templates for Qwen2.5-Math-7B models following, as shown in Table 4, and use Math-Verify (Hugging Face, 2025) as the verifier.

*Table 4.* Prompt template for Qwen2.5-Math-7B.

```
<|im_start|>system
You are a helpful assistant.<|im_end|>
<|im_start|>user
{input}
Please reason step by step, and put your final answer within \boxed{}.<|im_end|>
<|im_start|>assistant
```

## G. RL Training Configuration

For MATH, we use hyperparameters shown in Table 5, with a prompt batch size of 1,024 and 8 rollouts per prompt. The sampling temperature during training is set to 1.0, and maximum context length is set to 4,096 tokens. We update the model with a mini-batch size of 256 and learning rate of 1e-6. For DAPO-Math-17K, we use hyperparameters shown in Table 6, with a prompt batch size of 512 and 8 rollouts per prompt. The sampling temperature during training is set to 1.0, and maximum context length is set to 4,096 tokens. We update the model with a mini-batch size of 512 and learning rate of 1e-6.

*Table 5.* Hyperparameters for MATH

| Hyperparameter | Value |
|---|---|
| Optimizer | AdamW |
| Learning rate | $1e^{-6}$ |
| Batch size | 1024 |
| Samples per prompt | 8 |
| Mini-batch size | 256 |
| Max context length | 4K |
| Temperature | 1.0 |

*Table 6.* Hyperparameters for DAPO-Math-17K

| Hyperparameter | Value |
|---|---|
| Optimizer | AdamW |
| Learning rate | $1e^{-6}$ |
| Batch size | 512 |
| Samples per prompt | 8 |
| Mini-batch size | 512 |
| Max context length | 4K |
| Temperature | 1.0 |

# H. Detailed Description of Evaluation Benchmarks

In this section, we provide detailed descriptions of evaluation benchmarks used in our experiments. Our evaluation focuses on competition-level mathematical reasoning tasks that require multi-step logical reasoning and complex problem-solving abilities.

- **MATH / MATH-500** (Hendrycks et al., 2021) is a comprehensive benchmark designed to measure machine learning models' mathematical problem-solving abilities. It contains 12,500 challenging competition mathematics problems sourced from high school mathematics competitions such as AMC 10, AMC 12, AIME, and other renowned mathematical olympiads.

  **Dataset composition:**

    - **Total problems:** 12,500 (7,500 training + 5,000 test)
    - **MATH-500:** A curated subset of 500 representative test problems, commonly used for efficient evaluation

- **AIME** (MAA, 2025) is a prestigious examination following the AMC 10 and AMC 12 competitions, containing 15 problems with a 3-hour time limit. It serves as a qualifier for the USA Mathematical Olympiad (USAMO) and represents one of the most challenging standardized mathematical reasoning benchmarks.

  **Versions used:**

    - **AIME 2024:** 30 problems (2024 AIME I + AIME II)
    - **AIME 2025:** 30 problems (2025 AIME I + AIME II)

- **AMC** (MAA, 2023) is a series of mathematics competitions organized by the Mathematical Association of America (MAA). We specifically use AMC23 (40 problems), which includes problems from AMC 12.

# I. Detailed Formulation of Pass@$k$ Metric

When evaluating generative model performance, directly sampling $k$ results and checking for correct answers leads to high variance. To obtain more robust, unbiased evaluation results, we adopt the combinatorics-based expected estimation method proposed by Chen (2021). It generates $n$ samples ($n \geq k$) for each problem, counts the number of correct responses $c$, and computes the unbiased estimate of Pass@$k$ as:

$$Pass@k = \mathbb{E}_{x \sim \mathcal{D}} \left[ 1 - \frac{\binom{n-c}{k}}{\binom{n}{k}} \right]. \tag{22}$$

# J. Limitations

While SDAE demonstrates consistent improvements across multiple benchmarks, several limitations warrant discussion and point to promising directions for future research.

**Dependence on Correct Samples for Gravitational Source.** SDAE relies on the centroid of correct responses to serve as the gravitational source. For extremely challenging problems where few or no correct rollouts are sampled, the method must fall back to using the consensus of all responses (as discussed in Appendix B). While this fallback preserves some discriminative capacity between local and divergent errors, the exploration guidance may be less effective compared to

scenarios with well-defined positive centroids. Investigating adaptive strategies for gravitational source estimation under sparse correctness signals remains an open problem.

**Domain Generalization.** Our experiments focus exclusively on mathematical reasoning tasks, which benefit from deterministic verification and well-structured solution spaces. The effectiveness of SDAE on other reasoning domains, such as code generation, commonsense reasoning, or open-ended tasks where correctness is less binary, remains to be validated. In particular, tasks with subjective or multi-faceted evaluation criteria may require modified formulations of semantic gravity.

**Sensitivity to Internal Representation Quality.** SDAE relies on Latent Semantic Aggregation (LSA) to extract geometric information from the model's last-layer hidden states. If the base model's pre-trained embedding space is insufficiently developed or exhibits high noise in specific domains, the resulting semantic gravity may not accurately reflect the true reasoning diversity, potentially leading to biased advantage modulation.

