# OpenReview forum: "SDAE: Semantic-Diversity-Aware Exploration for Efficient Reinforcement Learning in Large Language Models"
_ICML.cc/2026/Conference — Submitted to ICML 2026_

### Official Review · Reviewer_e8Pk · 2026-03-05

**Soundness:** 2
**Presentation:** 2
**Significance:** 2
**Originality:** 2
**Overall Recommendation:** 3
**Confidence:** 4

**Summary:**

This paper introduces SDAE (Semantic-Diversity-Aware Exploration), a method for reinforcement learning with verifiable rewards (RLVR) that addresses diversity collapse in LLMs. SDAE leverages the geometric structure of responses in embedding space. It computes the centroid of correct samples and modulates each sample’s advantage based on its semantic distance to this centroid. Experiments on competition-level reasoning benchmarks show consistent improvements and reduced entropy collapse.

**Compliance With Llm Reviewing Policy:**

Affirmed.

**Final Justification:**

After two rounds of author rebuttals, I have updated my score to Weak Reject. While I recognize the paper’s potential, given the high acceptance bar of ICML and the additional experiments raised during rebuttal, I do not believe the work is yet ready for acceptance.

**Key Questions For Authors:**

- Can you provide evidence that the proposed semantic-gravity mechanism is fundamentally novel or necessary? There is a recent work focusing on a similar concept [2]. Please discuss the difference.
- How general and robust is the method beyond a single 7B math model?
- Can the authors provide quantitative/qualitative semantic diversity analyses to support the claim that centroid-based scaling meaningfully changes exploration dynamics?

[2] Nguyen et al. "Distance Is All You Need: Radial Dispersion for Uncertainty Estimation in Large Language Models." arXiv preprint arXiv:2512.04351 (2025).

**Limitations:**

No limitation section.

**Strengths And Weaknesses:**

Strength:
- The paper addressesa real issue in RLVR: diversity/entropy collapse.
- The “semantic centroid / semantic gravity” perspective provides an intuitive geometric interpretation of exploration in embedding space.
- The method shows improvements on math reasoning benchmarks such as MATH and AIME 2025, especially in Pass@k at large k.

Weakness:
- The novelty and technical contribution are not well justified. The paper does not compare or discuss alternatives that are simpler, such as explicit reward shaping that boosts semantically diverse samples (e.g., see [1]) and distance-based penalty scaling without centroid construction (e.g., the average of pairwise distances). It is unclear whether the centroid-based formulation is necessary or whether any function of semantic distance would work similarly.
- The experiments are not comprehensive. There is no test against KL-adaptive or trust-region variants with tuned entropy schedules. Also, it requires ablation comparing advantage modulation vs reward shaping. Thus, it is unclear whether the gains stem from the specific “semantic gravity” design or from introducing another diversity-sensitive scaling factor.
- All main experiments are conducted on Qwen2.5-Math-7B. There is no evaluation on different model families and sizes. This makes it difficult to assess generality.
- The claim that SDAE promotes semantic exploration beyond token-level entropy is not well-justified because, according to Fig. 5 and Fig. 6 b, other methods can have similar entropy as SDAE.

[1] Le et al. "Reasoning Under 1 Billion: Memory-Augmented Reinforcement Learning for Large Language Models." Transactions on Machine Learning Research 2025.

---

> ### Author Rebuttal · Authors · 2026-03-31
>
> Thank you for the thoughtful feedback. We address each concern below.
>
> > **W1:** No comparison with simpler alternatives (reward shaping, pairwise distance).
>
> **Response:**
> 1. **Reward shaping vs. advantage modulation:** Reward shaping modifies $r_i$ directly, altering the *optimization objective* and risking reward hacking. SDAE modulates the **advantage function**, preserving the original reward—changing *contribution strength*, not *what is optimized*.
> 2. **Pairwise vs. centroid:** Pairwise is O($N^2$); centroid is O($N$). More critically, pairwise measures dissimilarity to **all** responses, while centroid measures deviation from the **correct-solution consensus**. This directional reference enables the four-category framework, pairwise methods cannot distinguish "*correct but novel*" from "*incorrect but close*."
>
> Please see our response to **Reviewer hS78 Q3** for the full ablation results, where advantage modulation consistently outperforms reward shaping.
>
> > **W2:** No comparison with KL-adaptive/trust-region variants; need ablation of advantage modulation vs. reward shaping.
>
> Figure 5 compares SDAE with strong entropy-control baselines (Clip-Cov and KL-Cov). SDAE achieves **comparable entropy** (0.21 vs. 0.22 for Clip-Cov and 0.20 for KL-Cov) while improving accuracy, without per-token covariance overhead.
>
> Please see our response to **Reviewer hS78 Q3** for the ablation results.
>
> > **W3:** Only Qwen2.5-Math-7B; difficult to assess generality.
>
> We conducted **two new experiment sets** during rebuttal:
>
> (i) **Llama-3.1-8B-Instruct** (different model family), on MATH/AIME25/AMC23:
>
> | Method (MATH) | 1        | 16         | 64         | 256        |
> | ------------- | -------- | ---------- | ---------- | ---------- |
> | NSR           | _52.7_   | 77.4       | 84.2       | 88.6       |
> | W-REINFORCE   | 52.0     | 71.2       | 76.9       | 81.0       |
> | GRPO          | **54.1** | 76.7       | 83.1       | 87.4       |
> | GRPO w/ SDAE  | 52.3     | **_78.9_** | _**87.4**_ | _**90.1**_ |
>
> | Method (AIME25) | 1 | 16 | 64 | 256 |
> |---|---|---|---|---|
> | NSR | 0.2     | 3.0     | 9.8      | 26.7     |
> | W-REINFORCE     | 0.1     | 3.2     | 9.0      | 16.7 |
> | GRPO | 0.2 | 3.7 | 10.5 | 23.3 |
> | GRPO w/ SDAE | **0.3** | **4.4** | **14.2** | **33.3** |
>
> | Method (AMC23) | 1 | 16 | 64 | 256 |
> |---|---|---|---|---|
> | NSR | 28.7 | 65.9 | 76.9 | 82.5 |
> | W-REINFORCE | 27.8 | 49.0 | 58.5 | 70.0 |
> | GRPO | **29.1** | 65.1 | 76.7 | 82.5 |
> | GRPO w/ SDAE | 29.0 | **68.8** | **79.3** | **92.5** |
>
> (ii) **Qwen3-4B** (different scale, Non-Thinking Mode):
>
> | Method (MATH) | 1        | 16       | 32       | 64       |
> | ------------- | -------- | -------- | -------- | -------- |
> | NSR           | 93.3     | 96.6     | 96.8     | 97.1     |
> | W-REINFORCE   | 84.3     | 93.8     | 94.7     | 95.4     |
> | GRPO          | _93.9_   | _97.5_   | _97.9_   | _98.2_   |
> | GRPO w/ SDAE  | **94.7** | **98.0** | **98.3** | **98.6** |
>
> | Method (AIME25) | 1 | 16 | 32 | 64 |
> |---|---|---|---|---|
> | NSR | _58.4_ | _79.1_ | **83.0** | **86.7** |
> | W-REINFORCE | 23.5 | 53.0 | 58.7 | 63.3 |
> | GRPO | 56.8 | 76.4 | 78.9 | 80.0 |
> | GRPO w/ SDAE | **62.7** | **80.3** | 82.4 | 83.3 |
>
> | Method (AMC23) | 1 | 16 | 32 | 64 |
> |---|---|---|---|---|
> | NSR | _94.8_ | _99.8_ | 99.9 | **100.0** |
> | W-REINFORCE | 68.8 | 96.0 | 96.9 | 97.5 |
> | GRPO | 93.4 | 99.7 | **100.0** | **100.0** |
> | GRPO w/ SDAE | **96.6** | **100.0** | **100.0** | **100.0** |
>
> These results suggest that SDAE generalizes well across architectures and scales.
>
> > **W4:** Claim of semantic exploration beyond token-level entropy not well-justified; other methods achieve similar entropy.
>
> SDAE achieves **better downstream performance** (Tables 1&2) at **comparable entropy** to Clip-Cov/KL-Cov, suggesting gains are not merely from higher entropy. **Two policies can have identical entropy but entirely different exploration behaviors**, one from surface phrasing variations (token-level), another from genuinely different reasoning strategies (semantic-level). SDAE targets the latter, explaining superior performance at similar entropy.
>
> > **Q1:** Novelty of semantic gravity? Difference from RDS?
>
> RDS appeared on arXiv in Dec. 2025 and is **concurrent independent work** relative to ours.
>
> Key differences: (1) **Different domains**, RDS is inference-time uncertainty estimation; SDAE is training-time RL exploration guidance. (2) **Different centroids**, RDS uses all generations; SDAE uses only correct responses ($\mu^+$), enabling directional semantics. (3) **Four-category framework is SDAE's core contribution**, absent in RDS—RDS uses distance monotonically; SDAE cross-combines distance with correctness for opposing modulation effects. (4) In RDS distance is final output; in SDAE it is an intermediate modulation factor.
>
> > **Q2/Q3:** Generality? Semantic diversity analysis?
>
> See W3 response and our response to **Reviewer ci4D W4**, respectively.

---

> > ### Author Rebuttal · Reviewer_e8Pk · 2026-04-01
> >
> > Thank you for the responses and additional experiments. I appreciate the experiments with extra LLMs and the diversity analysis. However, key issues remain. Most critically, the comparison with reward shaping alternatives is insufficient. A table is included, but it is unclear which method was used. Given the pace of the field, comparison with recent GRPO reward shaping variants, such as length-based (DAPO), cosine, and memory-based approaches, is necessary, as these are simpler alternatives. On the pairwise baseline, I agree that centroid is more efficient, but a simple pairwise comparison remains easy to implement. With practical N=16, the computational difference could be small, so this comparison should be feasible. And for larger N, we can randomly sample pairs to reduce cost. For entropy-based baselines, I also recommend including recent competitive works [1, 2]. Overall, I think the paper in its current form is not ready for acceptance.
> >
> > [1] Shen, Han. "On entropy control in llm-rl algorithms." arXiv preprint arXiv:2509.03493 (2025).
> > [2] Jin, Renren, Pengzhi Gao, Yuqi Ren, Zhuowen Han, Tongxuan Zhang, Wuwei Huang, Wei Liu, Jian Luan, and Deyi Xiong. "Revisiting Entropy in Reinforcement Learning for Large Reasoning Models." arXiv preprint arXiv:2511.05993 (2025).

---

> > > ### Author Response · Authors · 2026-04-06
> > >
> > > Thank you for the additional clarification.
> > >
> > > > Reward-shaping baseline.
> > >
> > > The baseline in our response to **Reviewer hS78 Q3** is **GRPO + additive cosine reward shaping**:
> > >
> > > $r_i' = (2r_i - 1) \cdot (1 + SG_i)$,
> > >
> > > followed by standard GRPO advantage computation from the shaped rewards. This was designed as a controlled comparison under the same semantic signal. Under this setup, **advantage modulation** remained stronger than direct reward shaping (+6.6 Pass@256 on AIME25).
> > >
> > > > DAPO / cosine / memory-based methods.
> > >
> > > We agree these works are relevant, but they are not all direct one-to-one baselines for SDAE.
> > >
> > > - **DAPO** [1] is a full GRPO-family system, whose reward-shaping component mainly addresses **overlong/truncation reward noise** in long-CoT training.
> > > - The **cosine** [2] method is primarily **length-based reward shaping** for stabilizing CoT length growth.
> > > - The **memory-based** [3] method is more related in spirit, but it mainly targets **reward sparsity and weak exploration in tiny LLMs**, and relies on **external episodic memory and retrieval machinery**.
> > >
> > > We therefore view these methods as broadly related references rather than directly comparable baselines under the same problem setting.
> > >
> > > [1] DAPO: An Open-Source LLM Reinforcement Learning System at Scale, 2025
> > >
> > > [2] Demystifying Long Chain-of-Thought Reasoning in LLMs, 2025
> > >
> > > [3] Reasoning Under 1 Billion: Memory-Augmented Reinforcement Learning for Large Language Models, 2025
> > >
> > > > Pairwise baseline.
> > >
> > > We added a direct pairwise comparison using **Qwen2.5-Math-7B**, trained on **MATH** and evaluated on **AIME25**. The pairwise baseline uses the **mean pairwise distance** among sampled responses.
> > >
> > > |        Method (AIME25)         |  Pass@1  |  Pass@4  | Pass@16  | Pass@64  | Pass@256 |
> > > | :----------------------------: | :------: | :------: | :------: | :------: | :------: |
> > > | SDAE w/ **Pairwise Distances** |   10.1   |   18.1   |   25.6   |   34.5   |   56.7   |
> > > |  SDAE w/ **Centroid (Ours)**   | **11.7** | **20.0** | **30.0** | **44.1** | **63.3** |
> > >
> > > These results show that the pairwise baseline is meaningful, but the centroid-based design is clearly stronger. We believe this is because **pairwise averaging only measures overall dispersion, whereas the centroid provides a directional reference** anchored to the correct-solution set.
> > >
> > > > Recent entropy-control baselines.
> > >
> > > For ***AEnt*** [4], we completed a direct comparison under the same **Qwen2.5-Math-7B** backbone, trained on **MATH** and evaluated on **AIME25** and **AMC23**:
> > >
> > > | Method (AIME25) |  Pass@1  |  Pass@4  | Pass@16  | Pass@64  | Pass@256 |
> > > | :-------------: | :------: | :------: | :------: | :------: | :------: |
> > > |      AEnt       |   10.7   |   19.3   |   26.2   |   32.0   |   40.0   |
> > > | SDAE **(Ours)** | **11.7** | **20.0** | **30.0** | **44.1** | **63.3** |
> > >
> > > | Method (AMC23)  |  Pass@1  |  Pass@4  | Pass@16  | Pass@64  | Pass@256  |
> > > | :-------------: | :------: | :------: | :------: | :------: | :-------: |
> > > |      AEnt       | **63.7** |   75.8   |   84.3   |   93.0   |   97.5    |
> > > | SDAE **(Ours)** |   62.5   | **77.5** | **88.3** | **96.1** | **100.0** |
> > >
> > > SDAE clearly outperforms **AEnt** on **AIME25** and also surpasses it on **AMC23** at medium-to-large k.
> > >
> > > For ***Revisiting Entropy in Reinforcement Learning for Large Reasoning Models***, the paper does **not provide an open-source implementation**, so we report only an indirect comparison using the paper's reported **Pass@64** results. This comparison is relatively aligned in backbone and training data (**Qwen2.5-Math-7B** on **DAPO-Math-17K**), but still not perfectly matched because their setting uses **rollout.n=16** whereas ours uses **rollout.n=8**.
> > >
> > > |          Method (AIME24)          |  pass@64  |
> > > | :-------------------------------: | :-------: |
> > > |  Pos-Adv-Reweight (Stage-based)   |   56.67   |
> > > |   Pos-Adv-Reweight (Epoch-wise)   |   66.67   |
> > > | Pos-Adv-Reweight (Entropy-guided) | **73.33** |
> > > |            SDAE (Ours)            |   72.67   |
> > >
> > > |          Method (AIME25)          |  pass@64  |
> > > | :-------------------------------: | :-------: |
> > > |  Pos-Adv-Reweight (Stage-based)   |   46.67   |
> > > |   Pos-Adv-Reweight (Epoch-wise)   |   43.33   |
> > > | Pos-Adv-Reweight (Entropy-guided) |   40.00   |
> > > |            SDAE (Ours)            | **49.33** |
> > >
> > > |         Method (MATH500)          |  pass@64  |
> > > | :-------------------------------: | :-------: |
> > > |  Pos-Adv-Reweight (Stage-based)   |   96.00   |
> > > |   Pos-Adv-Reweight (Epoch-wise)   |   95.40   |
> > > | Pos-Adv-Reweight (Entropy-guided) |   95.40   |
> > > |            SDAE (Ours)            | **96.42** |
> > >
> > > Overall, we are very grateful for the reviewer's careful and constructive suggestions. They prompted us to add the most relevant missing comparisons and made the empirical picture of the paper substantially clearer. We hope this streamlined evidence helps clarify the paper's positioning, empirical value, and methodological boundaries.
> > >
> > > [4] On Entropy Control in LLM-RL Algorithms, 2025

---

### Official Review · Reviewer_o3h3 · 2026-03-13

**Soundness:** 2
**Presentation:** 4
**Significance:** 3
**Originality:** 3
**Overall Recommendation:** 4
**Confidence:** 4

**Summary:**

The paper proposes SDAE, which leverages the distance between each sampled rollout response and semantic gravity of positive samples to modulate advantage functions for RL training, so that the LLM is encouraged to produce diverse answers.

**Compliance With Llm Reviewing Policy:**

Affirmed.

**Final Justification:**

The supplementary experiments have largely addressed my confusion, and the perspective proposed in this paper also shows a certain degree of novelty. Therefore, I have decided to raise my score from weak-reject to weak-accept.

**Key Questions For Authors:**

- Q1: Why don’t you compare with other baselines mentioned in your related work, such as token-level entropy regularization or negative sample reinforcement, which also focus on increasing the diversity of the responses during RL training? It seems that only GRPO w/ Entropy Adv is a related baseline, and there is no obvious difference between SDAE and Entropy Adv.
- Q2: Will directly mean-pooling the hidden states of the last-layer loss a large amount of information in responses? Simply using a mean pool operation may not be enough to measure the semantics well. Have you tried some other semantic representation approaches, which may be helpful in improving the overall performance?
- Q3: Have you performed any repeated experiments? If so, why the standard deviation (std) results have not been reported to better demonstrate the performance gain of your method? As a single inference might not be sufficient to reach some of the results in Table 1 considering the total number of questions in the dataset.

**Limitations:**

L1: SADE relies on group-related RL algorithm like GRPO, as it need to calculate semantic gravity and modulate gravity advantage. This could restrict the method's applicability, making it unsuitable for direct use with a broader set of RL algorithms.

**Strengths And Weaknesses:**

**Strengths:**
- S1: The four-category semantic geometry framework that helps distinguish responses to assign more suitable reward is novel, and the usage of gravity provides new insights for better RL training.
- S2: This work aims to address the reasoning bottlenecks in the RL process and attributes them to diversity collapse. The research area is very important for the development of LLM’s reasoning ability, and the improvement provides valuable contributions to research.
- S3: The manuscript is clearly written and well structured, guiding readers smoothly from a broad overview down to specific implementation details. Besides, the work properly positions itself in the context of prior literature and clearly discuss the difference between SDAE and those exploration strategies in reinforcement learning.

**Weaknesses:**
- W1: Encouraging diversity in LLM responses does not necessarily help us obtain a better model. Helping the LLM to overcome the sub-optimal state during RL training is what really matters. Since reinforcement learning will quickly decrease the entropy of LLM to ensure that the model can get higher Pass@1 score and remember a strategy to produce correct answers for users. For users, more diverse responses among a group of rollouts have very limited value.
- W2: In most experiments, the gain from the SDAE method is not significant. Furthermore, the settings of the baselines are not sufficiently unified. All baseline methods should be compared under the **same** setting.

---

> ### Author Rebuttal · Authors · 2026-03-31
>
> Thank you for the thoughtful feedback. We address each concern below.
>
> > **W1:** Encouraging diversity doesn't necessarily yield a better model. RL should help overcome suboptimal states; diverse rollouts have limited user value.
>
> We believe it is important to distinguish a key concept: **SDAE encourages diversity during training to improve the final model, rather than providing diverse responses at inference time.**
>
> The critical evidence is that **SDAE significantly improves Pass@1**:
>
> - AIME25: Pass@1 is **+3.7%** higher than GRPO (Table 1)
> - AMC23: Pass@1 is **+12.8%** higher than GRPO (Table 1)
> - MATH500: Pass@1 is **+2.2%** higher than GRPO (Table 1)
>
> The mechanism is as follows: maintaining diverse exploration during training prevents the policy from falling into suboptimal local minima (as illustrated by the entropy collapse shown in Fig. 5). A model that explores diverse reasoning strategies during training ultimately learns a richer set of problem-solving skills, resulting in better greedy decoding (Pass@1) performance. This is precisely the **exploration-exploitation trade-off at the core of RL theory**.
>
> The improvements in Pass@k further indicate that SDAE-trained models possess a higher reasoning ceiling, which is also of significant value for inference-time scaling (e.g., best-of-N sampling, majority voting).
>
> > **W2:** Gains are not significant; baseline settings not unified.
>
> **Significance of gains:** All baselines are recent SOTA methods; improvements over them are inherently challenging. SDAE shows **substantial gains**:
> - Table 2 (AIME25): Pass@256 = **63.3%**, +**6.6%** over strongest baseline W-REINFORCE, +**13.3%** over vanilla GRPO. REINFORCE w/ SDAE reaches **66.7%**, breaking the base model ceiling (46.7%).
> - Table 1: vs. Entropy Adv, similar AIME25 Pass@256, but Pass@1 +**2.4%**, AMC23 Pass@1 +**9.9%**.
>
> **Unified settings:** Table 1 follows Cheng et al. [1]; Table 2 follows Zhu et al. [2]. SDAE trains under *exactly the same* hyperparameters and datasets as respective baselines.
>
> [1] Cheng, Daixuan, et al. "Reasoning with exploration: An entropy perspective." *Proceedings of the AAAI Conference on Artificial Intelligence*. Vol. 40. No. 36. 2026.
>
> [2] Zhu, Xinyu, et al. "The surprising effectiveness of negative reinforcement in llm reasoning." *arXiv preprint arXiv:2506.01347* (2025).
>
> > **Q1:** Why not compare with other baselines from related work? No obvious difference between SDAE and Entropy Adv.
>
> **In fact, these baselines are already included:** NSR (all benchmarks in Table 2, SDAE consistently outperforms), Entropy Adv (Table 1: SDAE Pass@1 +2.4%, AMC23 Pass@1 +9.9%), PSR and W-REINFORCE (Table 2).
>
> > **Q2:** Does mean-pooling lose significant information?
>
> SDAE only needs **distance metrics**, not full reconstruction, just whether two responses follow similar strategies.
>
> Table 3 compares LSA against external encoders (0.6B and 4B); external models did **not** **outperform** simple LSA.
>
> > **Q3:** Repeated experiments? Why no standard deviation reported?
>
> We agree that reporting variance is essential for verifying the reliability of results. To this end, we conducted independent repeated inference experiments for GRPO w/ SDAE on AIME25 using 5 different random seeds, with consistent evaluation settings (temperature=0.6, top-p=0.95). The results are as follows (mean ± std):
>
> |    Method    |       1       |       2       |       4       |       8       |      16       |      32       |      64       |      128      |      256      |
> | :----------: | :-----------: | :-----------: | :-----------: | :-----------: | :-----------: | :-----------: | :-----------: | :-----------: | :-----------: |
> |     NSR      |     10.0      |     14.6      |     19.2      |     24.1      |     29.3      |     34.6      |     40.2      |     46.0      |     53.3      |
> | W-REINFORCE  |     10.6      |     15.3      |     20.0      |     24.7      |     29.7      |     34.6      |     40.5      |     47.8      |     56.7      |
> |     GRPO     |     10.3      |     14.7      |     19.4      |     24.0      |     28.4      |     32.8      |     37.3      |     42.5      |     50.0      |
> | GRPO w/ SDAE | 11.6$\pm$0.56 | 15.8$\pm$0.24 | 20.4$\pm$0.75 | 24.9$\pm$0.49 | 30.2$\pm$0.22 | 36.5$\pm$0.25 | 44.0$\pm$0.16 | 53.7$\pm$0.64 | 63.3$\pm$2.37 |
>
> Small standard deviations indicate stability. The 13.3% Pass@256 gain is **5.6× the std**; even the conservative estimate (mean−std=60.9%) exceeds the strongest baseline by 4.2%, confirming gains are not sampling artifacts.
>
> > **L1:** SDAE relies on group-related RL algorithms, limiting applicability.
>
> Table 2 demonstrates SDAE works with **both GRPO and REINFORCE**—two fundamentally different policy gradient algorithms (REINFORCE w/ SDAE achieves 66.7% Pass@256). The only requirement is sampling multiple rollouts per prompt for centroid computation—**standard practice in modern RLVR pipelines**.

---

> > ### Author Rebuttal · Reviewer_o3h3 · 2026-04-04
> >
> > We thank the author for their detailed and thoughtful effort in rebuttal. Most questions are resolved and the effectiveness of SDAE is now better supported. Even the improvement of PASS@1 metrics is still not too large.

---

> > > ### Author Response · Authors · 2026-04-06
> > >
> > > We thank the reviewer for the further response. Regarding “*Even the improvement of PASS@1 metrics is still not too large,*” we would like to add a few brief clarifications.
> > >
> > > SDAE is designed to mitigate **entropy collapse / exploration collapse** in RL reasoning training, so the key signal is not only Pass@1, but also the **reasoning ceiling** under larger sampling budgets.
> > >
> > > **1. SDAE's gains are not limited to `Pass@1`; moreover, SDAE-based variants consistently outperform GRPO under different optimization frameworks, and this is especially clear on the more challenging AIME25 benchmark.**
> > >
> > > (i) **`Pass@k` gains of GRPO w/ SDAE over GRPO**
> > >
> > > | Dataset     | Pass@1 | Pass@2 | Pass@4 | Pass@8 | Pass@16 | Pass@32 | Pass@64 | Pass@128 | Pass@256 |
> > > | ----------- | -----: | -----: | -----: | -----: | ------: | ------: | ------: | -------: | -------: |
> > > | MATH gain   |   +0.8 |   +1.8 |   +1.5 |   +2.3 |    +1.5 |    +2.7 |    +2.3 |     +1.9 |     +1.7 |
> > > | AIME25 gain |   +1.4 |   +1.1 |   +0.6 |   +0.5 |    +1.6 |    +3.5 |    +6.8 |    +11.1 |    +13.3 |
> > > | AMC23 gain  |   +0.8 |   +1.9 |   +2.9 |   +3.5 |    +3.2 |    +2.9 |    +2.7 |     +3.0 |     +2.5 |
> > >
> > > (ii) **`Pass@k` gains of REINFORCE w/ SDAE over GRPO**
> > >
> > > | Dataset     | Pass@1 | Pass@2 | Pass@4 | Pass@8 | Pass@16 | Pass@32 | Pass@64 | Pass@128 | Pass@256 |
> > > | ----------- | -----: | -----: | -----: | -----: | ------: | ------: | ------: | -------: | -------: |
> > > | MATH gain   |   +4.7 |   +3.3 |   +3.7 |   +4.8 |    +7.2 |    +6.0 |    +5.1 |     +4.2 |     +3.7 |
> > > | AIME25 gain |   +2.9 |   +3.3 |   +3.0 |   +2.9 |    +3.5 |    +5.1 |    +7.9 |    +12.1 |    +16.7 |
> > > | AMC23 gain  |   -0.3 |   +1.4 |   +2.4 |   +2.8 |    +2.5 |    +2.5 |    +3.0 |     +3.4 |     +2.5 |
> > >
> > > These results show that SDAE does not merely provide a small Pass@1 improvement; rather, SDAE-based variants bring stable gains over GRPO under different optimization frameworks. **GRPO w/ SDAE** shows positive gains at every `k` on all three datasets. **REINFORCE w/ SDAE** also outperforms GRPO at every `k` on MATH and AIME25, while on AMC23, although `Pass@1` is only slightly lower by 0.3, the gains become positive from `Pass@2` onward and reach **+2.5** at `Pass@256`. On the more difficult AIME25 benchmark in particular, the gains expand clearly with `k`: GRPO w/ SDAE grows from **`Pass@1 +1.4`** to **`Pass@256 +13.3`**, and REINFORCE w/ SDAE from **`Pass@1 +2.9`** to **`Pass@256 +16.7`**.
> > >
> > > **2. In recent strong baselines, the `Pass@1` gains themselves are often not large either.**
> > >
> > > | Recent method                                                | Compared against        | Reported Pass@1 gain                        |
> > > | ------------------------------------------------------------ | ----------------------- | ------------------------------------------- |
> > > | Entropy Adv in *Reasoning with Exploration: An Entropy Perspective* (**AAAI 2026**; published on January 20, 2026) | GRPO on Qwen2.5-Math-7B    | AIME25 `+1.3`, AMC23 `+2.9`, MATH500 `+0.1` |
> > > | W-REINFORCE in *The Surprising Effectiveness of Negative Reinforcement in LLM Reasoning* (**NeurIPS 2025**; published on November 30, 2025) | GRPO on Qwen2.5-Math-7B | MATH `+0.3`, AIME25 `+0.3`, AMC23 `+0.3`    |
> > >
> > > In other words, for exploration-oriented RL methods, Pass@1 often improves only moderately, while the larger benefits appear in higher-`k` regions; what distinguishes SDAE is that it preserves this high-pass advantage while still delivering stable Pass@1 gains.
> > >
> > > **3. Even relative to recent strong baselines, SDAE still shows steady improvements.**
> > >
> > > Taking **W-REINFORCE** as an example, **GRPO w/ SDAE** further improves `Pass@1` by **+0.5 / +1.1 / +0.5** on **MATH / AIME25 / AMC23**, with corresponding `Pass@256` gains of **+0.5 / +6.6 / +2.5**. At the same time, **REINFORCE w/ SDAE** further improves `Pass@1` on **MATH / AIME25** by **+4.4 / +2.6**, with corresponding `Pass@256` gains of **+2.5 / +10.0**. This shows that SDAE's advantage is not limited to comparisons against vanilla GRPO, nor is it tied to a single RL backbone; it continues to deliver gains across different optimization frameworks, especially in the high-`k` regime.
> > >
> > > **4. In addition, we have also supplemented the recent new baselines raised by Reviewer e8Pk.**
> > >
> > > In our follow-up to **Reviewer e8Pk**, we have added comparisons and discussion on recent **entropy-control baselines**, and we refer the reviewer to that response for the detailed results. **We hope these additional results can help provide a more complete view of the paper's empirical value, and we sincerely thank the reviewer for the time and feedback.**

---

### Official Review · Reviewer_hS78 · 2026-03-13

**Soundness:** 3
**Presentation:** 2
**Significance:** 2
**Originality:** 2
**Overall Recommendation:** 4
**Confidence:** 4

**Summary:**

The authors propose a reinforcement learning exploration method based on semantic space geometry, which allows the model to not only pursue correctness during training, but also to more effectively retain and encourage diverse and innovative reasoning paths, thereby improving the reasoning ability and exploration ceiling of large models.

**Compliance With Llm Reviewing Policy:**

Affirmed.

**Final Justification:**

The problem addressed in this paper is indeed worth studying, and the authors have provided a thorough discussion in the rebuttal. However, the overall work is more like a summary of experimental observations and lacks rigorous theoretical support. Taking into account the questions raised by other reviewers as well as the authors’ responses, I ultimately conclude that this paper should receive a weak accept.

**Key Questions For Authors:**

1. Why can the last layer of hidden states in mean-pooling represent the semantics and reasoning path of the entire answer? This is a strong assumption and requires further proof.
2. Correctness and semantic distance are two orthogonal dimensions. Is there any quantitative evidence to prove that they are indeed approximately independent? The paper lacks derivation and experimental proof of this conclusion.
3. Phase 3: Four-Category Advantage Modulation. Eq5. Why is it superior to other distance functions or other shaping methods? Why is it better in terms of convergence, stability, or bias-variance? The paper lacks theoretical analysis of this advantage shaping method and needs to compare its effectiveness with other entropy-based advantage shaping methods to demonstrate its validity.
4. Does the CoT of the model's reasoning process truly become more accurate? Is there any qualitative or quantitative analysis regarding the accuracy of the reasoning process?

**Limitations:**

yes, the authors adequately discuss the limitations and potential negative societal impact of their work.

**Strengths And Weaknesses:**

Strengths: The authors modeled the distance of the answer from the center of the correct answer cluster in the semantic space to dynamically adjust the advantages during the training process, and the paper provides sufficient experimental verification.

Weaknesses: The experiments described in the paper are based on numerous assumptions, but these assumptions are not fully proven in the paper. For example:
1. Why are correct solutions farther from the centroid of the correct answer more novel and more deserving of a reward?
2. Why can the last layer of hidden states in mean-pooling represent the semantics and reasoning path of the entire answer?
3. Why are correctness and semantic distance two orthogonal dimensions?
These conclusions are novel, but they are more like experimental assumptions and are not sufficiently mathematically proven or discussed in the paper. For details, please refer to Key Questions For Authors.

---

> ### Author Rebuttal · Authors · 2026-03-31
>
> Thanks for your thoughtful feedback. We address your raised concerns as follows.
>
> > **W1:** Why are correct solutions farther from the centroid more novel and deserving of reward?
>
> As RL training progresses, the policy concentrates probability mass on a few dominant strategies; the centroid captures this "consensus." A correct response far from the centroid has arrived at the answer via a *different* reasoning path. Providing stronger rewards for such responses serves two purposes: **(1) prevent further collapse onto the dominant path, and (2) actively incentivize alternative strategies—precisely the mechanism to combat diversity collapse.** This parallels curiosity-driven exploration in classical RL, where centroid distance serves as a proxy for "semantic rarity."
>
> > **W2:** Why can last-layer hidden states with mean-pooling represent answer semantics?
>
> - **Theoretical support:** $G^2RL$ [1] proves that $Φ$ derives entirely from the last layer, establishing it as the unique "sensitivity bottleneck." LSA efficiently approximates this gradient feature.
> - **Empirical validation (Table 3):** LSA achieves 63.3% Pass@256 on AIME25, outperforming both the 4B external encoder (60.0%) and the 0.6B encoder (56.7%).
> - **Established NLP practice:** Last-layer mean-pooling is widely used in Sentence-BERT [2] and modern embedding models, retaining sufficient semantic signals for distance measurement.
>
> [1] Liang, Zhenwen, et al. "Can LLMs Guide Their Own Exploration? Gradient-Guided Reinforcement Learning for LLM Reasoning." *arXiv preprint arXiv:2512.15687* (2025).
>
> [2] Reimers, Nils, and Iryna Gurevych. "Sentence-bert: Sentence embeddings using siamese bert-networks." *Proceedings of the 2019 conference on empirical methods in natural language processing and the 9th international joint conference on natural language processing (EMNLP-IJCNLP)*. 2019.
>
> > **W3:** Why are correctness and semantic distance orthogonal dimensions?
>
> Thank you for raising this important point. We clarify that our use of "orthogonal" here is functional rather than strictly mathematical. What we mean is that correctness and semantic distance capture **complementary, non-redundant** information. We will adjust our wording in the revised paper.
>
> **Empirical correlation analysis:** To further support this point, we computed the point-biserial correlation between correctness and semantic gravity (SG) throughout training (Qwen2.5-Math-7B on MATH, 140 steps total):
>
> | Training Phase | $\rho$ |
> |---|:-:|
> | Early (0–50) | −0.0370 |
> | Mid (50–100) | 0.0516 |
> | Late (100+) | 0.0182 |
>
> Across all phases, the correlation remains very small in magnitude ($|\rho| < 0.1$), suggesting that correctness and semantic distance are only weakly correlated and therefore provide complementary signals rather than redundant ones.
>
> > **Q1/Q2:** Same as W2/W3.
>
> > **Q3:** Why is Eq.5 superior to other distance functions or shaping methods?
>
> The multiplicative design $A^{SDAE} = A·SG$ has three desirable properties: (1) **Sign preservation:** Since $SG ≥ 0$, correct responses remain rewarded and incorrect responses remain penalized. (2) **Automatic four-category behavior** without thresholds or conditional logic. (3) **No extra hyperparameters:** additive shaping ($A+λ·SG$) requires tuning $λ$; multiplicative form is naturally hyperparameter-free after batch normalization.
>
> We conducted ablation experiments on the MATH dataset (GRPO training on Qwen2.5-Math-7B, evaluated on AIME25), comparing:
>
> - (i) **Advantage modulation + cosine** (this paper, Eq. 5);
> - (ii) **Additive reward shaping + cosine**: $r_i' = (2r_i - 1) \cdot (1 + SG_i)$, then compute GRPO advantage from the shaped rewards;
> - (iii) **Advantage modulation + Euclidean distance**: replacing cosine distance with Euclidean distance.
>
> | Variant                                    |  Pass@1  | Pass@16  | Pass@64  | Pass@256 |
> | :----------------------------------------- | :------: | :------: | :------: | :------: |
> | GRPO (baseline)                            |   10.3   |   28.4   |   37.3   |   50.0   |
> | \+ Reward shaping                          |   10.9   |   28.7   |   40.2   |   56.7   |
> | \+ Advantage modulation + Euclidean        |   11.4   |   29.5   |   43.6   |   60.0   |
> | **+ Advantage modulation + cosine (ours)** | **11.7** | **30.0** | **44.1** | **63.3** |
>
> Advantage modulation significantly outperforms reward shaping (+6.6%), as it preserves the original reward structure. The small gap between distance metrics suggests that the main gains come from the **semantic geometric modulation mechanism itself**.
>
> > **Q4:** Does CoT reasoning become more accurate?
>
> See our response to **Reviewer ci4D W4**, which includes GPT-5.4-Thinking annotated step-level accuracy (**SDAE 55.8% vs. GRPO 47.9%**).

---

> > ### Author Rebuttal · Reviewer_hS78 · 2026-04-03
> >
> > My confusion has been initially resolved, and considering the content of the entire paper, I choose to keep my score.

---

> > > ### Author Response · Authors · 2026-04-06
> > >
> > > We thank the reviewer for the further update. We are glad that our previous response helped partially resolve the confusion.
> > >
> > > Since this is the final round, we would like to briefly emphasize that we have now added the following direct evidence addressing your main concerns:
> > >
> > > - Theoretical and empirical support for using mean-pooled last-layer representations.
> > > - Near-zero point-biserial correlations showing that correctness and SG capture complementary rather than redundant signals.
> > > - Controlled ablations showing that our advantage-modulation design outperforms additive reward shaping.
> > > - Distance-function ablations showing that, under the same advantage-modulation framework, the choice of distance metric has only a limited impact on the results.
> > > - Quantitative CoT-accuracy analysis showing improved reasoning quality over GRPO.
> > >
> > > In addition, following **Reviewer e8Pk**'s suggestion, we have also added comparisons to recent **entropy-control baselines** and additional ablations, and we refer the reviewer to our follow-up to **Reviewer e8Pk** for the detailed results.
> > >
> > > Taken together, we believe these additions substantially address the main concerns and make the paper's methodological motivation and empirical contribution clearer. We respectfully hope this strengthened evidence can support a more favorable overall reassessment of the paper.

---

### Official Review · Reviewer_ci4D · 2026-03-13

**Soundness:** 2
**Presentation:** 3
**Significance:** 2
**Originality:** 2
**Overall Recommendation:** 4
**Confidence:** 4

**Summary:**

RLVR suffers from diversity collapse. The author posits that this is because only the correctness signals are used regardless of solution diversity. To mitigate the "diversity collapse", the authors introduces SDAE. The authors constructs a centeroid from the last layer hidden states and assign rewards based on distance from the centroid. Diverse and good responses's advantage gets further amplified. Performance on competition math benchmarks show that the proposed method is effective in improving pass@1 and pass@k.

**Compliance With Llm Reviewing Policy:**

Affirmed.

**Key Questions For Authors:**

- Could the author show the length clipping ratio / reward curves for their experiments aside from just the entropy curves?

- Could the authors show qualitative examples on how their method distinguishes between semantic different solutions, and where existing embedding model fails?

**Limitations:**

Yes.

**Strengths And Weaknesses:**

Strengths:
 - The paper studies an important topic: diversity collapse in post-training.
 - The paper is easy to follow, although the font size in some of the figures were a bit small.
 - The reproducibility was good, the authors included many details and hyper-parameters in the paper. Very extensive.

Weaknesses:
 - The experiment setting is conducted under **max length = 4k** (Table 5 & 6 in Appendix G). This I believe is much smaller than usual settings. For example, DAPO paper [1] sets the max length to 16k. The paper "Does Reinforcement Learning Really Incentivize
Reasoning Capacity in LLMs Beyond the Base Model?" [2] sets the max length to 8k. With such a small max length, many of the generations are truncated, leading to unreliable results.

 - The paper only conducts experiments on a 2-year old model Qwen-2.5-7B-Instruct on two datasets: Math and DAPO. The generalizability of the method is questioned, I would encourage the authors to test on latest Qwen models (e.g. Qwen3/3.5) of a different size to test generalizability.

 - The paper's core method, is somewhat similar to EVOL-RL [3], which also uses an embedding model to determine diversity among responses and incorporate that to the reward. Similarly, Darling [4], also finetunes an embedding model for diversity evaluation. Given the similarity between the proposed SDAE and existing diversity encouraging RL methods, the paper does not too much meaningful findings to the community. It would be interesting if the authors could show that diversity in certain tasks (e.g. agentic / multi-turn) warrants a different way of evaluating.

 - (Minor) The authors claim that their method helps in solution variety, but there is no direct evaluation of this. It would be better if the authors could present a small set of questions where the answers are annotated by a frontier model about solution diversity. This would be direct evidence that the method improves solution diversity.

---

> ### Author Rebuttal · Authors · 2026-03-31
>
> Thank you for the detailed comments. Below, we clarify your key concerns:
>
> > **W1:** Max length=4k is much smaller than usual, causing truncation and unreliable results.
>
> Three clarifications: (1) **4K is Qwen2.5-Math-7B's native max context length**, not our arbitrary choice. The works you cited are based on different models that natively support longer contexts. (2) This setting directly follows our baselines Zhu et al. [1] and Cheng et al. [2] for fair comparison. (3) Qwen2.5-Math-7B is a non-long-CoT model whose typical responses fall within 4K—**truncation is not significant**. Clipping ratios and reward curves are in our anonymous repo [3].
>
> [1] Zhu, Xinyu, et al. "The surprising effectiveness of negative reinforcement in llm reasoning." *arXiv preprint arXiv:2506.01347* (2025).
>
> [2] Cheng, Daixuan, et al. "Reasoning with exploration: An entropy perspective." *Proceedings of the AAAI Conference on Artificial Intelligence*. Vol. 40. No. 36. 2026.
>
> [3] https://anonymous.4open.science/r/SDAE-Rebuttal-0E32/
>
> > **W2:** Only tested on Qwen-2.5-7B-Instruct; generalizability questioned.
>
> We added **two new experiment sets** during rebuttal. See our response to **Reviewer e8Pk W3** for complete results on **Llama-3.1-8B-Instruct** (different model family) and **Qwen3-4B** (different scale), where SDAE shows consistent improvements.
>
> > **W3:** Core method is similar to EVOL-RL and DARLING.
>
> Three **fundamental mechanistic differences**:
>
> **(1) Internal representations vs. external models.** EVOL-RL and DARLING rely on external models with per-rollout overhead. SDAE reuses the policy model's own hidden states with **no additional model inference**. This is theoretically supported by $G^2RL$ (the final layer is the unique optimization bottleneck) and empirically validated in Table 3, where LSA outperforms external encoders.
>
> **(2) Centroid geometry vs. pairwise/partitional diversity.** EVOL-RL computes O($N^2$) pairwise similarity; DARLING counts equivalence classes, both only measure "*how different responses are*." SDAE uses the correct-response **centroid** as reference, distinguishing valuable novel exploration from harmful divergence, impossible with pairwise methods.
>
> **(3) Differentiated error treatment.** EVOL-RL and DARLING treat errors nearly uniformly. SDAE uniquely distinguishes **local errors** (weak penalty) from **divergent errors** (strong penalty)—absent in both prior works.
>
> Additionally, EVOL-RL operates in an unlabeled setting (different problem domain), and SDAE requires no extra hyperparameters.
>
> > **W4:** No direct evaluation of solution diversity.
>
> We used **GPT-5.4-Thinking** as an annotator to analyze all 30 AIME25 problems (8 rollouts per problem).
>
> (i) **Quantitative results:**
>
> | Metric | GRPO | GRPO w/ SDAE |
> |---|---|---|
> | Avg. Correct Rollouts | 1.2 | 1.4 |
> | Avg. Distinct Strategy Clusters | 0.43 | 0.67 |
> | Strategy Diversity Ratio | 0.36 | 0.48 |
> | Avg. Reasoning Steps | 7.8 | 8.0 |
> | Step-Level Accuracy | 47.9% | 55.8% |
>
> SDAE produces **more semantically distinct correct strategies** and significantly higher intermediate reasoning accuracy.
>
> (ii) **Qualitative case (AIME25):** *Find the sum of all integer bases $b>9$ for which $17_b$ divides $97_b$.*
>
> | Method | Accuracy | Strategy Distribution |
> |---|---|---|
> | GRPO | 8/8 | 8×Strategy A, 0×Strategy B |
> | GRPO w/ SDAE | 8/8 | 5×Strategy A, 3×Strategy B |
>
> Both methods achieve 8/8 accuracy on this problem, but the strategy distributions are strikingly different:
>
> - **Strategy A: Divisor Enumeration**
>   Rewrite $17_b=b+7$ and $97_b=9b+7$, and observe $9b+7 = 9(b+7)-56$. Hence $b+7\mid 56$, and one can solve the problem by enumerating divisors of $56$.
>
> - **Strategy B: Quotient Parameterization**
>   Set $9b+7 = k(b+7)$, which gives $b(9-k)=7(k-1)$. Then solve the resulting integer constraints under $b>9$.
>
> GRPO collapses to a single strategy; SDAE consistently produces fundamentally different correct solutions.
>
> > **Q1:** Show clipping ratio / reward curves?
>
> We have uploaded them to our anonymous GitHub repository.
>
> > **Q2:** Qualitative examples of distinguishing solutions and where embedding models fail?
>
> For "find the sum of all positive integers n such that $n²+12n−2007$ is a perfect square," two correct solutions (difference-of-squares vs. discriminant method) have **large** cosine distance under LSA ($SG≈0.8953$), correctly reflecting different strategies, but **small** distance under ESE (**$SG≈0.1517$**), as external encoders focus on surface features rather than underlying reasoning structure.
>
> - **Solution A** (Difference of Squares): Set $n^2+12n-2007=m^2$. Complete the square: $(n+6)^2-m^2=2043$. So $(n+6-m)(n+6+m)=2043$ ... The sum is $1016+336+112=1464$.
>
> - **Solution B** (Discriminant): Treat $n^2+12n-(2007+m^2)=0$ as a quadratic in $n$. For integer solutions, the discriminant must be a perfect square: $\Delta=12^2+4(2007+m^2)$ ... This yields $n=1016,336,112$, and the sum is 1464.

---

> > ### Author Rebuttal · Reviewer_ci4D · 2026-04-01
> >
> > My concerns have been addressed, especially the 4k length. I have raised my score accordingly

---

> > > ### Author Response · Authors · 2026-04-01
> > >
> > > Thank you very much for raising your score and for your positive feedback. We greatly appreciate your careful reading of our rebuttal and are pleased that our clarification regarding the 4K length setting has resolved your concern. We will incorporate the above discussion into the revised version to make this point even clearer.
> > >
> > > Thank you again for your time and consideration.

---

### Decision · Program_Chairs · 2026-04-30

**Decision:**

Reject

**Comment:**

This paper was reviewed by four experts in the field. The reviewers agree that the paper addresses an important topic, namely diversity/entropy collapse in RL post-training. However, they raise substantial concerns regarding insufficient justification of novelty, noting similarities to existing diversity encouraging RL methods and questioning whether the centroid-based formulation is necessary. The reviewers indicate that experimental evaluation is not comprehensive, with reliance on Qwen2.5, limited datasets, small max length, and lack of comparisons to stronger baselines.
Considering the reviewers’ concerns, we regret that the paper cannot be recommended for acceptance at this time. The authors are encouraged to consider the reviewers’ comments when revising the paper for submission elsewhere.